# Adaptive Data Analysis for Growing Data

**Neil G. Marchant**
School of Computing & Information Systems
University of Melbourne, Australia
nmarchant@unimelb.edu.au

**Benjamin I. P. Rubinstein**
School of Computing & Information Systems
University of Melbourne, Australia
brubinstein@unimelb.edu.au

## Abstract

Reuse of data in adaptive workflows poses challenges regarding overfitting and the statistical validity of results. Previous work has demonstrated that interacting with data via differentially private algorithms can mitigate overfitting, achieving worst-case generalization guarantees with asymptotically optimal data requirements. However, such past work assumes data is *static* and cannot accommodate situations where data *grows* over time. In this paper we address this gap, presenting the first generalization bounds for adaptive analysis on dynamic data. We allow the analyst to adaptively *schedule* their queries conditioned on the current size of the data, in addition to previous queries and responses. We also incorporate time-varying empirical accuracy bounds and mechanisms, allowing for tighter guarantees as data accumulates. In a batched query setting, the asymptotic data requirements of our bound grows with the square-root of the number of adaptive queries, matching prior works' improvement over data splitting for the static setting. We instantiate our bound for statistical queries with the clipped Gaussian mechanism, where it empirically outperforms baselines composed from static bounds.

## 1 Introduction

The ubiquity of adaptive workflows in modern data science has raised concerns about the risk of overfitting and the validity of findings [1, 2]. In such adaptive workflows, data is reused over multiple steps, where the procedure or analysis at any given step may depend on results of previous steps. Common examples include hyperparameter tuning or model selection on a hold-out set [3–5], blending of exploratory and confirmatory data analysis [6, 7], and the reuse of benchmark/public datasets within a research community [8]. While adaptivity can enable more exploratory analysis, it is not covered by conventional guarantees of generalization and statistical validity, which assume the analysis is selected independently of the data [9].

A simple approach for enabling adaptive data analysis with generalization guarantees is to collect a fresh dataset whenever a step in the analysis depends on existing data. This can also be achieved by randomly splitting a dataset and using a separate split for each step. However, the data requirements of this approach may be prohibitive, scaling *linearly* in the number of adaptive steps. A line of work based on algorithmic stability [9–18] offers a significant improvement over data splitting, with data requirements that grow asymptotically with the *square-root* of the number of adaptive steps. The core result of this line of work is a *transfer theorem*, which guarantees that the outputs of an adaptive analysis are close to the expected outputs on the data distribution if (i) the analysis is stable under small changes to the dataset and (ii) the outputs are close to the empirical average on the dataset. *Differential privacy* [19] is commonly adopted as a notion of stability in this work, achieving the square-root dependence mentioned above, which is asymptotically optimal in the worst case [20, 21].

Most prior work on adaptive data analysis via algorithmic stability assumes a common setting, where the data is sampled i.i.d. from an unknown distribution and used by a mechanism to estimate an analyst's adaptive queries [9, 10, 15]. Generalization bounds are then obtained for a worst-case data

distribution and a worst-case analyst, who is actively trying to overfit. More recently, variations of this setting have been studied in an attempt to better reflect how data is used in practice [14, 18, 22, 23]. This includes replacing the assumption of i.i.d. data with weakly correlated data [23], or replacing a worst-case analyst by a dynamic model [22]. Concurrent work has also made progress on the former, providing generalization guarantees for correlated data by constraining the analyst to a class of "concentrated" queries [24].

A limitation of all prior work is the assumption that data is collected before the analysis begins, and remains *static* thereafter. However, it is common in practice for data to *grow* over time, and it may be undesirable to wait for all data to arrive or to ignore data that arrives after an analysis has begun. This *growing data* setting has been studied in the differential privacy literature, both for adaptive queries [25, 26] and for updating a fixed query whenever a dataset changes [27–30]. In this paper, we bridge the gap, obtaining the first generalization guarantees for adaptive data analysis (ADA) in the growing setting. We consider a fully adaptive analyst, who can determine not only the content of their queries, but also the timing and frequency of their submissions on-the-fly. This schedule can be conditioned on the current size of the data, as well as all past queries and responses.

To tackle the growing data setting, we introduce definitions and techniques that extend beyond existing ADA frameworks. A key innovation is our approach to bounding query error, which, following Jung et al. [15], incorporates a term comparing the query results evaluated on a posterior data distribution and the true data distribution. Our insight is that for the growing data setting, the posterior distribution must be marginalized over unseen future data at the time a query is submitted—a crucial departure from the static setting where the full dataset is known in advance. This yields a transfer theorem that depends on a corresponding variant of *posterior stability*. This dynamic nature of the posterior, whose support grows in a way that depends on the analyst's adaptive schedule, requires significant new analytical ideas to prove the conversion from differential privacy to posterior stability, which are instrumental in obtaining DP-based transfer theorems. This results in an additional factor in the error bound (compared to the static case for linear queries) proportional to the percentage increase in the dataset size.

We propose a non-uniform generalization of $(\epsilon, \delta)$-differential privacy where the $\delta$ parameter varies for each data point/time step, inspired by personalized privacy [31]. This permits us to obtain tighter generalization guarantees—the error bound increases as a function of the average $\delta$ over all time steps, rather than the maximum $\delta$ under the standard DP definition. These theoretical advances culminate in new bounds for various query types, including statistical queries, low-sensitivity queries, and low-sensitivity minimization queries using non-uniform differential privacy as a stability measure.

As a concrete application of our guarantees we consider using the clipped Gaussian mechanism to answer adaptive statistical queries. To ensure tight privacy accounting when the number of queries at each time step is chosen adaptively, we leverage a privacy filter [32] which supports fully adaptive composition. Our bound empirically outperforms baselines composed from bounds for static data. In a batched query setting, the asymptotic data requirements of our bound grow with the square-root of the number of adaptive queries for a fixed accuracy goal (assuming the ratio of final to initial data size is held constant). This improvement matches the improvement of bounds for static data [15] over the data splitting baseline.

## 2 Preliminaries

We introduce notation used throughout the paper. The sequence of integers from $n_1$ to $n_2$ inclusive is denoted by $[\![n_1, n_2]\!]$, or $[\![n_2]\!]$ when $n_1 = 1$. Given a sequence $\boldsymbol{x}$, we refer to the $t$-th element as $x_t$ and the length as $|\boldsymbol{x}|$. We use $\boldsymbol{x}_{[\![t_1, t_2]\!]}$ to denote the subsequence of $\boldsymbol{x}$ containing elements from index $t_1$ to $t_2$ inclusive, or $\boldsymbol{x}_{[\![t_2]\!]}$ when $t_1 = 1$. We use capital letters for random variables and lower case letters for realizations of a random variable. The uniform distribution over a set $\mathcal{S}$ is denoted $\mathcal{U}(\mathcal{S})$ and the normal distribution with mean $\mu$ and standard deviation $\sigma$ is denoted $\mathcal{N}(\mu, \sigma^2)$. The product distribution of $n$ i.i.d. random variables drawn from $\mathcal{D}$ is denoted $\mathcal{D}^n$.

### 2.1 Formulating Adaptive Data Analysis (ADA) for Growing Data

We propose a new formulation of adaptive data analysis for growing data that builds on prior work for static data [9, 10, 15].

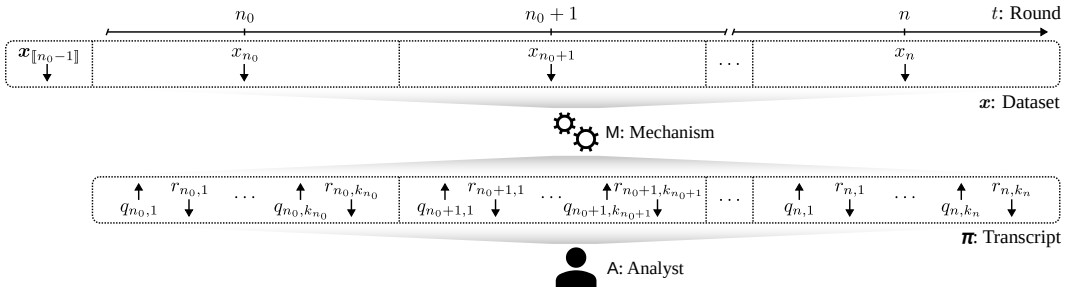

Figure 1: Schematic of our new setting for adaptive data analysis on growing data. The dataset is of size $n_0$ when the analysis begins, and grows by one data point in each round. The analyst asks queries adaptively in each round based on past responses, and receives a response from the mechanism before selecting the next query. The framework reduces to the static data setting when $n = n_0$.

**Dataset.** Let $\mathcal{P}$ be an unknown data distribution over a finite domain $\mathcal{X}$. We consider a growing dataset $\boldsymbol{X} = (X_1, X_2, \ldots)$, where each data point $X_t$ is drawn i.i.d. from $\mathcal{P}$, and the data points are indexed in order of arrival. We define the *snapshot* of $\boldsymbol{X}$ at index $t$, to be the portion of the data realized by index $t$, namely $\boldsymbol{X}_{[\![t]\!]}$. We study datasets over a fixed horizon $n$ so that $|\boldsymbol{X}| = n$.

**Analyst and mechanism.** We consider an analyst A who would like to estimate queries about the data distribution $\mathcal{P}$ *online* using the growing dataset $\boldsymbol{X}$. The analyst asks queries from a fixed query class $\mathcal{Q}$, such as the class of *statistical queries* (see Section 2.2). The analyst is prohibited from accessing $\boldsymbol{X}$ directly, but can instead submit queries to an online mechanism M that returns estimates using the current snapshot of $\boldsymbol{X}$. We assume M produces *feasible estimates*, meaning estimates are guaranteed to fall within the range of the query. This can be achieved by design [33] or by modifying M to project infeasible estimates onto the range of the query.

**Interaction.** Algorithm 1 specifies how the analyst interacts with the mechanism. To begin, the analyst selects an initial dataset size $n_0 \leq n$ and waits for the mechanism to receive $n_0$ data points (lines 1 and 4). The analyst then submits queries to the mechanism over multiple rounds, where each round is marked by the receipt of a new data point (lines 3–10). The rounds are indexed starting at $n_0$ so that index $t$ coincides with the size of the growing dataset. The number of queries asked in a given round $t$ is determined adaptively by the analyst (line 5). Since the analyst adaptively controls when each round

| **Algorithm 1** Interaction between A and M |
|---|
| 1: Wait for M to receive data $\boldsymbol{X}_{[\![n_0-1]\!]}$ |
| 2: Initialize empty transcript $\Pi$ |
| 3: **for** Round $t \in [\![n_0, n]\!]$ **do** |
| 4:      Wait for M to receive next data point $X_t$ |
| 5:      **while** Analyst not finished **do** |
| 6:          Generate query: $Q \sim \mathsf{A}(\Pi)$ |
| 7:          Estimate query response: $R \sim \mathsf{M}(Q; \boldsymbol{X}_{[\![t]\!]})$ |
| 8:          Append $(Q, R)$ to $\Pi_t$ in-place |
| 9:      **end while** |
| 10: **end for** |
| 11: **return** Transcript $\Pi$ |

terminates, they can force the mechanism to use a stale snapshot of the dataset $\boldsymbol{X}_{[\![t]\!]}$ even if newer data points (with indices $> t$) have arrived and are waiting to be ingested by the mechanism.

**Transcript.** The interaction yields a transcript $\Pi$ of the queries submitted in each round and the estimates produced by the mechanism (lines 2 and 8). The transcript is structured as a sequence of sequences $\Pi = (\Pi_1, \Pi_2, \ldots, \Pi_n)$ where $\Pi_t = ((Q_{t,1}, R_{t,1}), \ldots, (Q_{t,k_t}, R_{t,k_t}))$ records the query-estimate pairs from round $t$ in the order they were submitted. We denote the space of possible transcripts by $\mathcal{T} = \bigcup_{\boldsymbol{k} \in S(n,k)} \prod_{t=1}^{n} (\mathcal{Q} \times \mathcal{R})^{k_t}$, where $\mathcal{Q}$ is the query class, $\mathcal{R}$ is the range of the queries and $S(n,k) = \{\boldsymbol{k} \in [\![k]\!]^n : (\forall t < n_0)(k_t = 0) \wedge \sum_{t=1}^{n} k_t = k\}$ is the set of possible allocations of $k$ queries across $n$ rounds.[1]

Looking ahead, we will be interested in the stability of the transcript under perturbations to the dataset. It is therefore convenient to interpret the interaction in Algorithm 1 as a random map $\mathsf{I}(\boldsymbol{X}; \mathsf{A}, \mathsf{M})$ that takes a growing dataset $\boldsymbol{X} \in \mathcal{X}^n$ as input and returns a transcript $\Pi \in \mathcal{T}$ as output. We view A and M as parameters of $\mathsf{I}$, and drop the dependence on them where it is clear from context.

---

[1]In the definition of $S(n,k)$, the first statement in the predicate accounts for the fact that the analyst does not begin submitting queries until round $n_0$.

## 2.2 Query Classes

Following prior work [10, 15], we consider three query classes. We use $q(\boldsymbol{x}_{[\![t]\!]})$ to denote the result of a query $q \in \mathcal{Q}$ evaluated on a snapshot $\boldsymbol{x}_{[\![t]\!]}$ and $q(\mathcal{D})$ to denote the result evaluated on a data distribution $\mathcal{D}$.

**Low-sensitivity queries** are defined by a $\boldsymbol{\Delta}$-sensitive function $q : \mathcal{X}^* \to [0, 1]$ that maps a data snapshot to a scalar on the unit interval. We say $q$ is $\boldsymbol{\Delta}$-sensitive given $\boldsymbol{\Delta} = (\Delta_1, \dots, \Delta_n) \in \mathbb{R}_+^n$, if for all $t \in [\![n]\!]$ we have $|q(\boldsymbol{x}_{[\![t]\!]}) - q(\tilde{\boldsymbol{x}}_{[\![t]\!]})| \leq \Delta_t$ for any pair of neighboring snapshots $\boldsymbol{x}_{[\![t]\!]}, \tilde{\boldsymbol{x}}_{[\![t]\!]} \in \mathcal{X}^t$ that differ on one data point. The result of the query when evaluated on a data distribution $\mathcal{D}$ is $q(\mathcal{D}) := \mathbb{E}_{\boldsymbol{X} \sim \mathcal{D}}[q(\boldsymbol{X})]$.

**Statistical queries** are a subset of $\boldsymbol{\Delta}$-sensitive queries where each query is of the form $q(\boldsymbol{x}_{[\![t]\!]}) = \sum_{\tau=1}^{t} \tilde{q}(x_\tau)/t$ for some function $\tilde{q} : \mathcal{X} \to [0, 1]$. Since each query is fully specified by $\tilde{q}$, we refer to $\tilde{q}$ as $q$ when there is no ambiguity. The sensitivity satisfies $\Delta_t \leq 1/t$ for all $t \in [\![n]\!]$.

**Minimization queries** are solutions to parameter optimization problems defined by a data-dependent loss function. Due to space constraints, we discuss them in Appendix A.

## 2.3 Generalization and Stability

Algorithm 1 may fail to generalize if the mechanism leaks detailed information about the dataset that is exploited by the analyst when selecting queries. The degree of leakage is related to the stability of the interaction under perturbations to the dataset. Roughly speaking, a more stable interaction leaks less information and is more likely to generalize. In Section 3, we will derive generalization guarantees that depend on the stability of the interaction. In preparation, we now define how generalization and stability will be measured. For clarity of exposition, we focus on low-sensitivity and statistical queries here, and extend to minimization queries in Appendix A.

Consider the mechanism's response $R$ to query $Q$ in round $t$, for which the "true" answer to the query is the expected value on the data distribution, denoted $Q(\mathcal{P}^t)$. We measure generalization of $R$ in terms of the absolute difference $|R - Q(\mathcal{P}^t)|$, which we refer to as the *distributional error*. Our generalization guarantee for the analysis as a whole, takes the form of a high probability bound on the worst-case distributional error that holds jointly over all rounds, as defined below. Note that we consider bounds on the error $\alpha_t$ that vary as a function of the round index $t$, which permits the bound to improve as the dataset grows.

**Definition 2.1.** Let $\alpha_t \geq 0$ for all $t \in [\![n_0, n]\!]$ and $\beta \geq 0$. A mechanism M is $(\{\alpha_t\}, \beta)$-*distributionally accurate* if with probability $1 - \beta$ over the randomness in the dataset $\boldsymbol{X} \sim \mathcal{P}^n$ and transcript $\Pi \sim \mathsf{I}(\boldsymbol{X}; \mathsf{A}, \mathsf{M})$, the largest distributional error in the $t$-th round satisfies $\max_{(Q,R) \in \Pi_t} |R - Q(\mathcal{P}^t)| \leq \alpha_t$, and this holds jointly for all $t \in [\![n_0, n]\!]$, for any analyst A and any data distribution $\mathcal{P}$.

When deriving distributional accuracy bounds in the next section, we make use of a related accuracy bound that compares the mechanism's responses to raw empirical estimates evaluated on the current data snapshot. Consider again the mechanism's response $R$ to query $Q$ in round $t$, where the raw estimate using snapshot $\boldsymbol{X}_{[\![t]\!]}$ is denoted $Q(\boldsymbol{X}_{[\![t]\!]})$. We define the *snapshot error* of $R$ to be the absolute difference $|R - Q(\boldsymbol{X}_{[\![t]\!]})|$.[2] By analogy with Definition 2.1, we then define the following accuracy bound using snapshot error.

**Definition 2.2.** Let $\alpha_t \geq 0$ for all $t \in [\![n_0, n]\!]$ and $\beta \geq 0$. A mechanism M is $(\{\alpha_t\}, \beta)$-*snapshot accurate*[3] if with probability $1 - \beta$ over the randomness in the dataset $\boldsymbol{X} \sim \mathcal{P}^n$ and transcript $\Pi \sim \mathsf{I}(\boldsymbol{X}; \mathsf{A}, \mathsf{M})$, the largest snapshot error in the $t$-th round satisfies $\max_{(Q,R) \in \Pi_t} |R - Q(\boldsymbol{X}_{[\![t]\!]})| \leq \alpha_t$, and this holds for all $t \in [\![n_0, n]\!]$, for any analyst A and any data distribution $\mathcal{P}$.

As previously mentioned, our generalization guarantees depend on the stability of the interaction. We adapt the notion of *posterior stability* introduced by Jung et al. [15] to the growing data setting. For a query $Q$ in round $t$, it measures stability in terms of the absolute difference between the "true" answer

---

[2]$R$ and $Q(\boldsymbol{X}_{[\![t]\!]})$ do not generally coincide, since the mechanism may inject noise in its estimates.

[3]Cummings et al. [25] adopt a similar definition of accuracy for a DP mechanism operating on a growing dataset. However their definition assumes a non-adaptive analyst and holds for a worst-case dataset, whereas ours holds for a worst-case adaptive analyst assuming the growing dataset is drawn from $\mathcal{P}^n$.

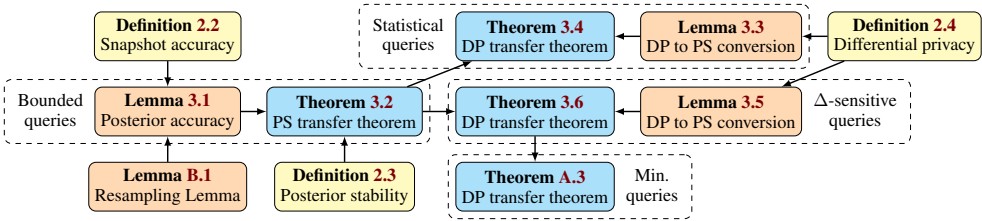

Figure 2: Outline of results in Section 3. Arrows indicate key dependencies, and dashed boxes indicate results that hold for a particular class of queries.

evaluated on the data distribution $\mathcal{P}^t$, and the answer evaluated on the posterior data distribution $\mathcal{Q}_\Pi^t := \mathcal{P}^t \mid \Pi$. As in the static setting, the posterior data distribution is conditioned on the full transcript $\Pi$ at the end of the interaction, however unlike the static setting, the distribution is only taken over the data available up to round $t$.

**Definition 2.3** ([15])**.** Let $\epsilon, \delta \geq 0$. An interaction $\mathsf{I}(\,\cdot\,;\,\cdot\,,\mathsf{M})$ is $(\epsilon, \delta)$-*posterior stable*, or $(\epsilon, \delta)$-PS for short, if with probability $1 - \delta$ over the randomness in the dataset $\boldsymbol{X} \sim \mathcal{P}^n$ and transcript $\Pi \sim \mathsf{I}(\boldsymbol{X}; \mathsf{A}, \mathsf{M})$, we have $\max_{(Q,R) \in \Pi_t} |Q(\mathcal{P}^t) - Q(\mathcal{Q}_\Pi^t)| \leq \epsilon$ for all $t \in [\![n_0, n]\!]$, and this holds for any analyst $\mathsf{A}$ and any data distribution $\mathcal{P}$.

We will show that posterior stability follows from *differential privacy* [19]. We consider a variation of the standard definition of differential privacy, where the level of privacy/stability varies non-uniformly over data points. Similar definitions have been used in dynamic data settings [34, 35] to facilitate tighter privacy accounting. We have the same motivation here—by bounding the $\delta$ parameter for each data point separately, we can obtain tighter generalization bounds.

**Definition 2.4.** Let $\epsilon \geq 0$ and $\boldsymbol{\delta} \colon [\![n]\!] \to [0, 1]$. An interaction $\mathsf{I}(\,\cdot\,;\,\cdot\,,\mathsf{M})$ is $(\epsilon, \boldsymbol{\delta})$-*differentially private*, or $(\epsilon, \boldsymbol{\delta})$-DP for short, if for all analysts $\mathsf{A}$, all rounds $t \in [\![n]\!]$, all pairs of neighboring growing datasets $(\boldsymbol{x}, \boldsymbol{x}') \in \mathcal{N}_t$ differing on the $t$-th data point, and all measurable events $E \subseteq \mathcal{T}$:

$$\mathbb{P}(\mathsf{I}(\boldsymbol{x}; \mathsf{A}, \mathsf{M}) \in E) \leq \mathsf{e}^\epsilon \, \mathbb{P}(\mathsf{I}(\boldsymbol{x}'; \mathsf{A}, \mathsf{M}) \in E) + \boldsymbol{\delta}(t).$$

We consider a *bounded* neighboring relation, meaning that $\mathcal{N}_t$ contains pairs of datasets $(\boldsymbol{x}, \boldsymbol{x}')$ where $\boldsymbol{x}$ can be obtained from $\boldsymbol{x}'$ by replacing the $t$-th data point. We note that this non-uniform privacy definition includes the standard (uniform) definition as a special case: in particular, $(\epsilon, \boldsymbol{\delta})$-DP implies $(\epsilon, \max_t \boldsymbol{\delta}(t))$-DP. We refer the reader to Appendix D for discussion and results on non-uniform DP.

As a stability measure for adaptive data analysis, DP has several advantages. First, DP can be interpreted as a privacy guarantee, not merely a bound on stability. Second, DP has been widely studied in statistics and machine learning for almost two decades, so there are established private mechanisms for common data analysis and learning tasks. Third, DP supports composition which simplifies privacy accounting of the interaction. For example, one could instantiate a privacy filter [32, 36] for a differentially private query-answering mechanism with target privacy parameters $\epsilon$ and $\delta$. This then allows for adaptive selection of the analyst's queries, the mechanism's algorithm, and the privacy parameters for individual queries, noting that the interaction may be forced to terminate early to ensure it satisfies $(\epsilon, \delta)$-DP. The post-processing property of DP is essential for this accounting to work, as the analyst's selection of the next query is viewed as post-processing on the mechanism's private estimates to previous queries. As a result, the analyst does not accrue an additional cost to privacy/stability, even in the worst case.

## 3 Generalization Guarantees for ADA on Growing Data

In this section, we present generalization guarantees for ADA in the growing data setting. Figure 2 provides an outline of our key results, where the generalization guarantees (a.k.a. transfer theorems) are shaded in blue. Due to space constraints, we present selected results here and defer proofs and technical results to Appendix B.

Recall that our aim is to obtain generalization guarantees in the form of high probability bounds on worst-case distributional accuracy. Our proof technique is based on Jung et al. [15], whose bounds

for static data outperform prior work [9, 10]. The core idea of the proof involves decomposing the distributional error into two terms using the triangle inequality:

$$|r - q(\mathcal{P}^t)| \leq |r - q(\mathcal{Q}_\pi^t)| + |q(\mathcal{Q}_\pi^t) - q(\mathcal{P}^t)|. \qquad (1)$$

Rather than comparing the response $r$ to query $q$ with the true value $q(\mathcal{P}^t)$ directly, we instead compare $r$ and $q(\mathcal{P}^t)$ with an intermediate value $q(\mathcal{Q}_\pi^t)$, which is the expectation of the query on the posterior distribution of the snapshot at round $t$ conditioned on the final transcript $\pi$. This intermediate value is chosen to align with the definitions of snapshot accuracy and posterior stability, which we have adapted for the growing data setting.

We obtain worst-case probabilistic bounds on each term in (1) separately: a bound on the first term follows indirectly from snapshot accuracy and a bound on the second term follows directly from posterior stability. The bound we obtain for the first term is stated below.

**Lemma 3.1.** *Suppose* M *is* $(\{\alpha_t\}, \beta)$*-snapshot accurate for* $[0, 1]$*-bounded queries. Then for any* $c > 0$*, with probability* $1 - \frac{\beta}{c}$ *with respect to the randomness in the dataset* $\boldsymbol{X} \sim \mathcal{P}^n$ *and transcript* $\Pi \sim \mathsf{I}(\boldsymbol{X}; \mathsf{A}, \mathsf{M})$*, we have for all* $t \in [\![n_0, n]\!]$ *that* $\max_{(Q,R)\in\Pi_t} |R - Q(\mathcal{Q}_\Pi^t)| \leq \alpha_t + c$*.*

The proof relies on an elementary observation stated in Lemma B.1 of Appendix B: that the joint distribution on datasets and transcripts does not change when the entire dataset is resampled from the posterior distribution $\mathcal{Q}_\Pi^n$ in the final round $n$. At first glance, this observation may not seem useful, as the expectation of the query in (1) is taken with respect to the posterior distribution of the data available at round $t$ when the query was submitted $\mathcal{Q}_\Pi^t$, not the posterior distribution of the entire dataset $\mathcal{Q}_\Pi^n$. However it turns out this is not a problem: Since the event of interest does not depend on data points received after the round $t^\star$ when the worst-case deviation from the posterior expectation occurs, we can apply Lemma B.1 and marginalize over the remaining data points.

Combining Lemma 3.1 with posterior stability yields our first generalization guarantee.

**Theorem 3.2** (PS transfer theorem). *Suppose* M *is an* $(\{\alpha_t\}, \beta)$*-snapshot accurate mechanism for* $[0, 1]$*-bounded queries and* $\mathsf{I}(\cdot; \cdot, \mathsf{M})$ *is an* $(\epsilon, \delta)$*-posterior stable interaction. Then for every* $c > 0$*,* M *is* $(\{\alpha_t'\}, \beta')$*-distributionally accurate for* $\alpha_t' = \alpha_t + c + \epsilon$ *and* $\beta' = \frac{\beta}{c} + \delta$*.*

When the error bounds $\{\alpha_t\}$ are constant, we recover the same bound as Jung et al. [15, Theorem 4], albeit for the more general growing data setting.

We now turn to deriving generalization guarantees using differential privacy as a measure of stability. These guarantees are derived by first converting differential privacy (DP) to posterior stability (PS) in a way that exploits the structure of the query class, and then invoking Theorem 3.2. These steps are visualized in Figure 2, where the DP to PS conversion results in Lemmas 3.3 and 3.5 lead to generalization guarantees in Theorems 3.4 and 3.6.

We begin with a conversion result for statistical queries.

**Lemma 3.3.** *An* $(\epsilon, \boldsymbol{\delta})$*-DP interaction* $\mathsf{I}(\cdot; \cdot, \mathsf{M})$ *for statistical queries is* $(\epsilon', \delta')$*-PS for* $\epsilon' = \mathsf{e}^\epsilon - 1 + 2c\sum_{t=1}^n \boldsymbol{\delta}(t)/n_0$*,* $\delta' = 1/c$ *and any* $c > 0$*.*

If $\epsilon$ and $c$ are both constant, then the scaling of the lower bound $\epsilon'$ as a function of the final dataset size $n$ depends on the functional form of $\sum_{t=1}^n \boldsymbol{\delta}(t)$. In the worst case where $\boldsymbol{\delta}(t)$ is uniform, we see that $\epsilon'$ grows linearly in $n$. In the static setting, where $n = n_0$ and $\boldsymbol{\delta}(t) = \delta$, this factor disappears and we recover the result of Jung et al. [15, Lemma 7]. Combining this lemma with Theorem 3.2 yields a generalization guarantee for statistical queries.

**Theorem 3.4.** *Suppose* M *is an* $(\{\alpha_t\}, \beta)$*-snapshot accurate mechanism and* $\mathsf{I}(\cdot; \cdot, \mathsf{M})$ *is an* $(\epsilon, \boldsymbol{\delta})$*-DP interaction for statistical queries. Then for any constants* $c, d > 0$*,* M *is* $(\alpha_t', \beta')$*-distributionally accurate for* $\alpha_t' = \alpha_t + \mathsf{e}^\epsilon - 1 + 2c\sum_{t=1}^n \boldsymbol{\delta}(t)/n_0 + d$ *and* $\beta' = \beta/d + 1/c$*.*

Next we consider low-sensitivity queries, where we obtain the following DP to PS conversion result.

**Lemma 3.5.** *An* $(\epsilon, \boldsymbol{\delta})$*-DP interaction* $\mathsf{I}(\cdot; \cdot, \mathsf{M})$ *for* $\boldsymbol{\Delta}$*-sensitive queries is* $(\epsilon', \delta')$*-posterior stable for* $\epsilon' = \mathsf{e}^\epsilon \max_{\tau_1 \in [\![n_0, n]\!]} \tau_1 \Delta_{\tau_1} - \min_{\tau_2 \in [\![n_0, n]\!]} \tau_2 \Delta_{\tau_2} + 4c \left(\sum_{t=1}^n \boldsymbol{\delta}(t)\right) \max_{\tau_3 \in [\![n_0, n]\!]} \Delta_{\tau_3}$*,* $\delta' = 1/c$ *and any* $c > 0$*.*

We see that the lower bound on the posterior stability depends on the extreme values of the sensitivity and the sensitivity weighted by dataset size. It is interesting to apply this lemma to statistical queries,

which are a subset of $\boldsymbol{\Delta}$-sensitive queries with $\Delta_t \leq 1/t$. We find $\epsilon' = \mathsf{e}^\epsilon - 1 + 4c \sum_{t=1}^n \boldsymbol{\delta}(t)/n_0$, which is looser than Lemma 3.3 by a factor of 2 in the last term. We again point out that the bound reduces to Jung et al. [15, Lemma 15] in the static case, where $n = n_0$ and $\boldsymbol{\delta}(t) = \delta$. By combining this lemma with Theorem 3.2 we obtain a generalization guarantee for low-sensitivity queries.

**Theorem 3.6.** *Suppose* $\mathsf{M}$ *is an* $(\{\alpha_t\}, \beta)$*-snapshot accurate mechanism and* $\mathsf{I}(\,\cdot\,;\,\cdot\,, \mathsf{M})$ *is an* $(\epsilon, \boldsymbol{\delta})$*-DP interaction for* $\boldsymbol{\Delta}$*-sensitive queries. Then for any constants* $c, d > 0$, $\mathsf{M}$ *is* $(\{\alpha_t'\}, \beta')$*-distributionally accurate for* $\alpha_t' = \alpha_t + \mathsf{e}^\epsilon \max_{\tau_1 \in [\![n_0, n]\!]} \tau_1 \Delta_{\tau_1} - \min_{\tau_2 \in [\![n_0, n]\!]} \tau_2 \Delta_{\tau_2} + 4c \left( \sum_{t=1}^n \boldsymbol{\delta}(t) \right) \max_{\tau_3 \in [\![n_0, n]\!]} \Delta_{\tau_3} + d$ *and* $\beta' = \beta/d + 1/c$.

Finally, we provide a generalization guarantee for minimization queries in Theorem A.3 of Appendix A.

# 4 Application: Gaussian Mechanism

In this section, we instantiate our generalization guarantees for statistical queries using the Gaussian mechanism. We focus on the Gaussian mechanism since it is simple to describe, it is easily adapted for growing data, and it is known to be optimal for answering a small number of queries $k \ll n^2$ on a static dataset of size $n$ [10].[4] For ease of exposition, we assume the number of queries asked in each round is fixed before the analysis begins, while the queries themselves are adaptively chosen. This simplifies the privacy accounting and makes for a more direct comparison with prior work in the static setting [15]. In Appendix C.2, we remove this assumption, presenting a guarantee for the more general setting where the analyst adaptively decides how many queries to ask in each round. This guarantee relies on fully adaptive composition via a privacy filter [32]. All proofs for this section can be found in Appendix C.

## 4.1 Generalization Guarantee

We begin by defining the Gaussian mechanism for growing data. We also include a clipped variant that produces feasible estimates to $[0,1]$-bounded queries (see Section 2.1).

**Definition 4.1.** The *Gaussian mechanism* perturbs an estimate to a query $q$ based on snapshot $\boldsymbol{x}_t$ by adding Gaussian noise with round-dependent standard deviation $\sigma_t > 0$. Specifically, we have $\mathsf{M}(q; \boldsymbol{x}_t) = q(\boldsymbol{x}_t) + z$ with $z \sim \mathcal{N}(0, \sigma_t^2)$. The *clipped Gaussian mechanism* composes the Gaussian mechanism with the function $\mathrm{clip}_{[0,1]}(x) = \max(0, \min(x, 1))$ as a post-processing step.

We now analyze privacy for the clipped Gaussian mechanism. The result below is obtained using zero-concentrated differential privacy (zCDP, see Definition D.3), as it provides sharp composition bounds for the Gaussian mechanism [37]. After proving that the interaction satisfies $\rho$-zCDP, we convert to $(\epsilon, \boldsymbol{\delta})$-DP using Corollary D.9, which generalizes a result of Canonne, Kamath, and Steinke [38, Corollary 13] to non-uniform privacy parameters.

**Lemma 4.2.** *Consider an interaction* $\mathsf{I}(\cdot; \cdot, \mathsf{M})$ *where* $\mathsf{M}$ *is the ordinary or clipped Gaussian mechanism. Suppose the analyst decides to submit* $k_\tau$ *statistical queries in round* $\tau \in [\![n_0, n]\!]$ *before* $\mathsf{I}(\cdot; \cdot, \mathsf{M})$ *is executed. Then* $\mathsf{I}(\cdot; \cdot, \mathsf{M})$ *satisfies* $(\epsilon, \boldsymbol{\delta})$*-DP for any* $\epsilon > 0$ *and* $\boldsymbol{\delta}(t) \leq \psi(\gamma^\star, \boldsymbol{\rho}(t), \epsilon)$, *where* $\boldsymbol{\rho}(t) = \sum_{\tau=n_0}^n k_\tau \mathbf{1}_{[t \leq \tau]}/2\sigma_\tau^2 \tau^2$, $\psi(\gamma, \rho, \epsilon) = \mathsf{e}^{(\gamma-1)(\gamma \rho - \epsilon)} \left( 1 - \gamma^{-1} \right)^\gamma /(\gamma - 1)$ *and* $\gamma^\star = \arg\min_{\gamma \in (1,\infty)} \psi\left( \gamma, \max_{t \in [\![n]\!]} \boldsymbol{\rho}(t), \epsilon \right)$.

Next we analyze snapshot accuracy. The bound depends on the inverse CDF of the Gaussian distribution, which is related to the inverse complementary error function $\mathrm{erfc}^{-1}$.

**Lemma 4.3.** *For any* $\beta \in (0, 1)$, *the clipped Gaussian mechanism with* $\sigma_t \propto \alpha_t$ *is* $(\{\alpha_t\}, \beta)$*-snapshot accurate for* $k$ *queries with*

$$\frac{\alpha_t}{\sqrt{2}\sigma_t} = \mathrm{erfc}^{-1}\left( 2 - 2\left( 1 - \frac{\beta}{2} \right)^{\frac{1}{k}} \right) < \mathrm{erfc}^{-1}\left( \frac{\beta}{k} \right).$$

Combining Lemmas 4.2 and 4.3 with Theorem 3.4 yields the following generalization guarantee.

---

[4]In the regime where $k \gg n^2$, a variant of the private multiplicative weights mechanism for growing data can be used instead [25].

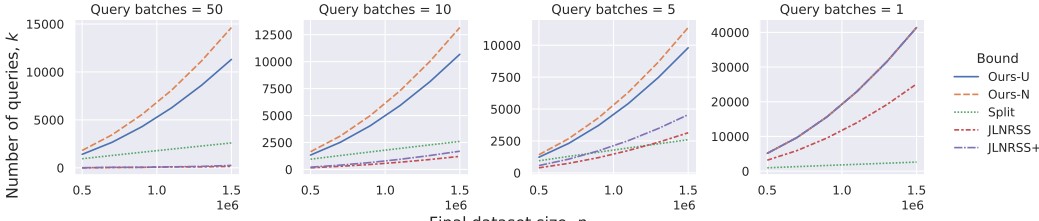

Figure 3: Comparison of the number of adaptive statistical queries that can be answered with error tolerance $\alpha = 0.1$ and uniform coverage probability $1 - \beta = 0.95$ using a growing dataset with growth ratio $n/n_0 = 3$ in a batched query setting. The number of queries (vertical axis) is plotted as a function of the final dataset size $n$ (horizontal axis), bound (curve style) and the number of query batches $b$ (horizontal panel). The right-most panel ($b = 1$), represents the static baseline setting where the analyst forgoes intermediate responses and submits all queries only after the entire dataset of size $n$ has arrived.

**Theorem 4.4.** *Suppose the conditions of Lemma 4.2 hold and assume $\sigma_t = \sigma > 0$. Then the clipped Gaussian mechanism is $(\alpha', \beta')$-distributionally accurate for $k$ statistical queries for any $\beta' \in (0, 1)$ and $\alpha' = \min_{\sigma, \beta, \epsilon \in \Theta} \lambda(\sigma, \beta, \epsilon)$, where*

$$\lambda(\sigma, \beta, \epsilon) = \sqrt{2}\sigma \, \mathrm{erfc}^{-1}\left(\frac{\beta}{k}\right) + \mathsf{e}^{\epsilon} - 1 + \frac{\beta}{\beta'} + \frac{2\sum_{\tau=1}^{n} \boldsymbol{\delta}(\tau)}{n_0 \beta'} + \frac{2}{\beta'}\sqrt{\frac{2\beta \sum_{\tau=1}^{n} \boldsymbol{\delta}(\tau)}{n_0}},$$

$\Theta = \{(\sigma, \beta, \epsilon) \in \mathbb{R}^3 : \sigma > 0, 0 < \beta < 1, \epsilon \geq 0\}$ *and* $\boldsymbol{\delta}(\cdot)$ *is defined in Lemma 4.2.*

### 4.2 Empirical Comparison with Alternative Guarantees

We empirically compare our generalization bounds for growing data with baselines composed from bounds for static data. When instantiating the bounds, we must specify how many queries $k_t$ are submitted in each round $t$.[5] For simplicity, we assume $k$ queries are evenly split into $b$ batches, with a batch being submitted every $T = (n - n_0)/b$ rounds.[6] Concretely, for $b > 1$ we assume $k_t = k/b$ for $t \in \{n_0, n_0 + T, n_0 + 2T, \ldots, n\}$ and $k_t = 0$ otherwise. This setting represents a middle ground between two extremes: where all $k$ queries are submitted in the initial round or final round. To provide a strong baseline from prior work, we treat the $b = 1$ case specially: it represents the static data setting, where the analyst waits for all $n$ data points to arrive before submitting all $k$ queries in a single batch at the final round $t = n$.

We consider the following generalization bounds:

- **Ours-N**. Theorem 4.4 with non-uniform privacy.

- **Ours-U**. Theorem 4.4 with uniform privacy (expression $\boldsymbol{\delta}(t)$ is replaced by $\max_{\tau \in [\![n]\!]} \boldsymbol{\delta}(\tau)$ in our bounds).

- **JLNRSS**. Proposition C.4 in Appendix C.3. Composes the static bound of Jung et al. [15, Theorem 13] over query batches using a fresh static dataset for each query batch, thereby yielding a worst-case guarantee over all queries. The static bound depends on parameters that are optimized under the constraints $\beta = \delta$ and $c = d$.

- **JLNRSS+**. A tighter variant of **JLNRSS** that differs in two aspects: (1) the parameters are optimized without imposing the simplifying constraints mentioned above, and (2) the conversion from zCDP to $(\epsilon, \delta)$-DP is based on the tighter result that we use (Corollary D.9).

- **Split**. An analogue of the sampling splitting baseline from prior work [10, 15] adapted for growing data. It splits incoming data points into samples of size $\lfloor n/k \rfloor$ and answers each query using a fresh sample. Unlike the other methods, this method may respond with a delay if a query arrives before a fresh sample is ready. Since there is no data reuse, a generalization bound follows directly from Hoeffding's bound and the union bound (see Appendix C.4).

---

[5]Since the bounds are worst case with respect to the analyst and data distribution, no simulation is necessary.
[6]When division by $b$ yields a remainder $r$, we distributed $r$ evenly across the first $r$ batches/rounds.

Figure 3 plots the number of adaptive queries $k$ that can be answered as a function of the final dataset size $n$ while guaranteeing a confidence interval around estimates of $\alpha' = 0.1$ with uniform coverage probability $1 - \beta' = 0.95$, following the empirical settings of Jung et al. [15]. When varying $n$, we set the initial dataset size $n_0 = n/3$ to maintain a constant growth ratio. This choice models a practical scenario where a substantial dataset is available before adaptive analysis begins, which is necessary to obtain non-vacuous guarantees in a fully adaptive setting. To present the tightest possible guarantees for each method, we follow the approach of Jung et al. [15] and numerically optimize over the free parameters of the bounds (e.g., $\sigma, \beta, \epsilon$ for **Ours-N** and **Ours-U**) for each point plotted. Our primary goal is to compare generalization guarantees under data reuse, so the resulting privacy parameters may vary slightly between points. We note, however, that the optimal values were found to be stable in practice, varying only beyond the first significant figure (e.g., $\sigma \approx 0.008$, $\beta \approx 10^{-5}$, $\epsilon \approx 0.04$).

We observe quadratic growth in $n$ for **Ours-U** and **Ours-N**, linear growth in $n$ for **Split** and slower quadratic growth in $n$ for **JLNRSS** and **JNLRSS+**. In this regime, **Ours-N** generally outperforms the other bounds, which is not surprising as **JLNRSS** in particular is not optimized for growing data $b > 1$. We provide additional results in Appendix E examining the error for a fixed number of queries; the effect of $b$; and a setting where $n_0$ is fixed and the growth ratio varies.

## 5 Related Work

Various methods have been proposed in the statistics community for adaptive analysis of static data. For instance, $\alpha$-investing and related methods can be used to control false discovery rates for sequential hypothesis testing [39, 40]. Another line of work aims to ensure statistical validity when model selection and significance testing are performed on the same dataset [41–43]. However, these methods are specialized and place restrictions on the analyst [9].

Our paper builds on a body of work exploiting a connection between stable algorithms and generalization for adaptive data analysis [5, 9–18]. Dwork et al. [9] were first to establish a *transfer theorem* showing that differentially private algorithms are sufficient to guarantee high-probability bounds on the worst-case error of an adaptive analysis. In subsequent work [10, 15], simpler proofs of the transfer theorem were given that achieve sharper bounds, while covering a broader range of queries. Recent work has obtained bounds that improve on the worst case, by conditioning on data/queries [11, 14, 18] or constraining the analyst [22]. In a similar spirit, concurrent work also constrains the analyst's queries, which in turn allows them to provide guarantees for certain classes of correlated, non-i.i.d. data [24]. Lower bounds have also been studied exploiting connections to cryptography [20, 21, 44]. However, all of this prior work is limited to static data.

While there is no prior work on adaptive analysis of dynamic data, this setting has been studied in differential privacy. One line of work is known as differential privacy *under continual observation* [27, 28], where the goal is to repeatedly estimate a fixed function of a dataset whenever new data arrives. An elementary task in this setting is estimating the number of ones in a binary stream, which can be solved using the binary mechanism [27, 28]. More recently, alternative mechanisms have been proposed that achieve tighter error bounds [29, 30] while also maintaining computational efficiency [45]. These counting mechanisms have been used as a primitive to tackle other tasks including frequency estimation [46, 47], learning [29, 48] and graph spectrum analysis [49]. A second line of work studies differential privacy for more general kinds of adaptive queries on dynamic data [25, 26]. Cummings et al. [25] design mechanisms for growing data that call black-box mechanisms for static data on a schedule. Qiu and Yi [26] go beyond growing data, designing mechanisms that estimate adaptive linear queries on datasets where items may be inserted or deleted over time. Our paper provides generalization guarantees for many of these mechanisms.

Several works have obtained generalization bounds for adaptive data analysis that do not rely on differential privacy. While differential privacy yields accuracy bounds that hold with high probability, one can also study weaker bounds that hold on average via connections to information theory [10, 50, 51]. Concepts from computational learning theory, such as Rademacher complexity, have also been used to obtain data-dependent generalization bounds for adaptive testing [52]. However, estimating data-dependent bounds on Rademacher complexity may be computationally challenging.

Connections have been made between adaptive data analysis and seemingly disparate areas. Steinke, Nasr, and Jagielski [53] develop a method for privacy auditing that runs the algorithm under audit once on a single dataset rather than many times on adjacent datasets. The analysis of their method

relies on generalization bounds for adaptive data analysis tailored for uniformly distributed binary data. Liu et al. [54] use static and dynamic analysis to estimate the adaptivity of a program to assist in bounding its generalization error.

# 6  Conclusion

This paper extends the current understanding of generalization in adaptive data analysis to dynamic scenarios where data arrives incrementally over time, a setting increasingly relevant in many data-driven fields. Our approach builds on and extends the tightest known worst-case generalization guarantees for static data [15] by incorporating time-varying accuracy bounds and addressing the additional complexity introduced by data growth. Compared with bounds for static data, our bounds incorporate an additional factor proportional to the data growth, associated with the stability of the analysis as measured by differential privacy. We instantiate our bounds for three query classes and demonstrate an empirical improvement over baselines for adaptive statistical queries answered with the clipped Gaussian mechanism.

There are various opportunities to extend our work. While it is conventional to assume i.i.d. data when studying generalization, as we have done here, it would be interesting to consider non-i.i.d. growing data where the data distribution evolves over time. This may require bounds that depend on the rate of evolution, akin to bounds that depend on the correlation for non-i.i.d. data in the static setting [23]. Another practical direction is to tighten our (mostly) worst-case bounds by conditioning on the actual queries and data realized in the analysis. This could be informed by similar work for the static setting [11, 14, 18]. Finally, we believe there are opportunities to design new differentially private mechanisms for different kinds of adaptive queries on dynamic data, as there has been limited work in this area to date [25, 26].

## Acknowledgments and Disclosure of Funding

We acknowledge support from the Australian Research Council Discovery Project DP220102269.

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

# A   Results for Minimization Queries

In this appendix, we provide a generalization guarantee for the class of low-sensitivity minimization queries. Queries in this class are parameterized by a loss function $L : \mathcal{X}^* \times \Theta \to [0, 1]$, where the first argument is a data snapshot and the second argument is a set of parameters from a parameter space $\Theta$. We require that $L$ is $\boldsymbol{\Delta}$-sensitive in its first argument, meaning for all $t \in [\![n]\!]$ and all $\theta \in \Theta$ we have $|L(\boldsymbol{x}_{[\![t]\!]}, \theta) - L(\boldsymbol{x}'_{[\![t]\!]}, \theta)| \leq \Delta_t$ for any pair of neighboring snapshots $\boldsymbol{x}_{[\![t]\!]}, \boldsymbol{x}'_{[\![t]\!]} \in \mathcal{X}^t$. When evaluated on a snapshot, the result of the query is $q(\boldsymbol{x}_{[\![t]\!]}) \in \arg\min_{\theta \in \Theta} L(\boldsymbol{x}_{[\![t]\!]}, \theta)$. When evaluated on a data distribution $\mathcal{D}$, the result is $q(\mathcal{D}) \in \arg\min_{\theta \in \Theta} \mathbb{E}_{\boldsymbol{X} \sim \mathcal{D}}[L(\boldsymbol{X}, \theta)]$.

The definitions of distributional and snapshot accuracy that appear in Section 2.3 must be adapted for minimization queries. Minimization queries require a different treatment because the output of the query is not scalar-valued in general, but rather a set of parameters $\theta \in \Theta$. We use the loss function $L$ associated with the query to measure the distributional and snapshot accuracy as defined below.

**Definition A.1.** Let $\alpha_t \geq 0$ for all rounds $t \in [\![n_0, n]\!]$ and $\beta \geq 0$. A mechanism M is $(\{\alpha_t\}, \beta)$-*distributionally accurate* if for all analysts A and data distributions $\mathcal{P}$

$$\mathbb{P}_{\boldsymbol{X} \sim \mathcal{P}^n, \Pi \sim \mathsf{I}(\boldsymbol{X}; \mathsf{A}, \mathsf{M})} \left( \bigcup_{t=n_0}^n \bigcup_{(L,\theta) \in \Pi_t} \left\{ \left| \mathbb{E}_{\tilde{\boldsymbol{X}} \sim \mathcal{P}^t}[L(\tilde{\boldsymbol{X}}, \theta)] - \min_{\tilde{\theta} \in \Theta} \mathbb{E}_{\tilde{\boldsymbol{X}} \sim \mathcal{P}^t}[L(\tilde{\boldsymbol{X}}, \tilde{\theta})] \right| \geq \alpha_t \right\} \right) \leq \beta.$$

**Definition A.2.** Let $\alpha_t \geq 0$ for all rounds $t \in [\![n_0, n]\!]$ and $\beta \geq 0$. A mechanism M is $(\{\alpha_t\}, \beta)$-*snapshot accurate* if for all analysts A and data distributions $\mathcal{P}$

$$\mathbb{P}_{\boldsymbol{X} \sim \mathcal{P}^n, \Pi \sim \mathsf{I}(\boldsymbol{X}; \mathsf{A}, \mathsf{M})} \left( \bigcup_{t=n_0}^n \bigcup_{(L,\theta) \in \Pi_t} \left\{ \left| L(\boldsymbol{X}_{[\![t]\!]}, \theta) - \min_{\tilde{\theta} \in \Theta} L(\boldsymbol{X}_{[\![t]\!]}, \tilde{\theta}) \right| \geq \alpha_t \right\} \right) \leq \beta.$$

Following prior work [10, 15], we obtain a generalization guarantee for $\boldsymbol{\Delta}$-sensitive minimization queries by applying Theorem 3.6 to a related set of $2\boldsymbol{\Delta}$-sensitive scalar-valued queries.

**Theorem A.3.** *Suppose M is an $(\{\alpha_t\}, \beta)$-snapshot accurate mechanism and $\mathsf{I}(\,\cdot\,; \,\cdot\,, \mathsf{M})$ is an $(\epsilon, \boldsymbol{\delta})$-DP interaction for $\boldsymbol{\Delta}$-sensitive minimization queries. Then for any constants $c, d > 0$, M is $(\{\alpha'_t\}, \beta')$-distributionally accurate for $\alpha'_t = \alpha_t + 2e^\epsilon \max_{\tau_1 \in [\![n_0, n]\!]} \tau_1 \Delta_{\tau_1} - 2 \min_{\tau_2 \in [\![n_0, n]\!]} \tau_2 \Delta_{\tau_2} + 8c \sum_{t=1}^n \boldsymbol{\delta}(t) \max_{\tau_3 \in [\![n_0, n]\!]} \Delta_{\tau_3} + d$ and $\beta' = \beta/d + 1/c$.*

*Proof.* We begin by defining a mapping $f : \mathcal{Q}_{\min} \times \Theta \to \mathcal{Q}_{\mathrm{ls}} \times [0, 1]$ that takes a $\boldsymbol{\Delta}$-sensitive minimization query-estimate pair $(L, W) \in \mathcal{Q}_{\min} \times \Theta$ and returns a $2\boldsymbol{\Delta}$-sensitive scalar-valued query-estimate pair $(q, r) \in \mathcal{Q}_{\mathrm{ls}} \times [0, 1]$:

$$f(L, W) = (q, r) \quad \text{with} \quad q(\boldsymbol{x}) := L(\boldsymbol{x}, W) - \min_{W' \in \Theta} L(\boldsymbol{x}, W) \quad \text{and} \quad r := 0.$$

Since $f$ does not depend on the dataset, we can apply it to all pairs in the minimization query transcript $\Pi$ to yield a transformed transcript $\Pi'$ that also satisfies $(\epsilon, \boldsymbol{\delta})$-DP by the post-processing guarantee (Theorem D.6). It is straightforward to see that the transformed transcript $\Pi'$ is $(\{\alpha_t\}, \beta)$-snapshot accurate iff the original transcript $\Pi$ is.

Next, observe that the probability of interest can be upper-bounded using Jensen's inequality to swap the order of the minimization and expectation operations:

$$\mathop{\mathbb{P}}_{\boldsymbol{X} \sim \mathcal{P}^n, \Pi \sim \mathsf{I}(\boldsymbol{X})} \left( \bigcup_{t=n_0}^{n} \bigcup_{(L,W) \in \Pi_t} \left\{ \left| \mathop{\mathbb{E}}_{\boldsymbol{X}' \sim \mathcal{P}^t} [L(\boldsymbol{X}', W)] - \min_{W' \in \Theta} \mathop{\mathbb{E}}_{\boldsymbol{X}' \sim \mathcal{P}^t} [L(\boldsymbol{X}', W')] \right| > \alpha_t' \right\} \right)$$

$$\leq \mathop{\mathbb{P}}_{\boldsymbol{X} \sim \mathcal{P}^n, \Pi \sim \mathsf{I}(\boldsymbol{X})} \left( \bigcup_{t=n_0}^{n} \bigcup_{(L,W) \in \Pi_t} \left\{ \left| \mathop{\mathbb{E}}_{\boldsymbol{X}' \sim \mathcal{P}^t} \left[ L(\boldsymbol{X}', W) - \min_{W' \in \Theta} L(\boldsymbol{X}', W') \right] \right| > \alpha_t' \right\} \right)$$

$$= \mathop{\mathbb{P}}_{\boldsymbol{X} \sim \mathcal{P}^n, \Pi \sim \mathsf{I}(\boldsymbol{X})} \left( \bigcup_{t=n_0}^{n} \bigcup_{(q,r) \in \Pi_t'} \left\{ \left| r - \mathop{\mathbb{E}}_{\boldsymbol{X}' \sim \mathcal{P}^t} [q(\boldsymbol{X}')] \right| > \alpha_t' \right\} \right).$$

In the last line above, we have rewritten the event in terms of the transformed transcript $\Pi'$. Applying Theorem 3.6 upper bounds this probability by $\beta'$, completing the proof. $\qquad\square$

## B    Proofs for Section 3

**Lemma B.1** (Resampling Lemma [15])**.** *Let $E \subseteq \mathcal{X}^n \times \mathcal{T}$ be any event. Then*
$$\mathop{\mathbb{P}}_{\boldsymbol{X} \sim \mathcal{P}^n, \Pi \sim \mathsf{I}(\boldsymbol{X})} ((\boldsymbol{X}, \Pi) \in E) = \mathop{\mathbb{P}}_{\boldsymbol{X} \sim \mathcal{P}^n, \Pi \sim \mathsf{I}(\boldsymbol{X}), \boldsymbol{X}' \sim \mathcal{Q}_\Pi} ((\boldsymbol{X}', \Pi) \in E).$$

*Proof.* The result follows by writing the probabilities as expectations, invoking the definition of $\mathcal{Q}_\Pi$, and using the fact that $\boldsymbol{x}$ and $\boldsymbol{x}'$ can be swapped without changing the expectation:

$$\mathop{\mathbb{P}}_{\boldsymbol{X} \sim \mathcal{P}^n, \Pi \sim \mathsf{I}(\boldsymbol{X}), \boldsymbol{X}' \sim \mathcal{Q}_\Pi} ((\boldsymbol{X}', \Pi) \in E) = \sum_{\boldsymbol{x}, \pi, \boldsymbol{x}'} \mathbb{P}(\boldsymbol{X} = \boldsymbol{x}, \Pi = \pi) \mathop{\mathbb{P}}_{\boldsymbol{X}' \sim \mathcal{Q}_\pi} (\boldsymbol{X}' = \boldsymbol{x}') \mathbf{1}_{[(\boldsymbol{x}', \pi) \in E]}$$

$$= \sum_{\pi, \boldsymbol{x}'} \mathbb{P}(\Pi = \pi) \mathbb{P}(\boldsymbol{X} = \boldsymbol{x}' \mid \Pi = \pi) \mathbf{1}_{[(\boldsymbol{x}', \pi) \in E]}$$

$$= \sum_{\boldsymbol{x}, \pi} \mathbb{P}(\boldsymbol{X} = \boldsymbol{x}, \Pi = \pi) \mathbf{1}_{[(\boldsymbol{x}, \pi) \in E]}$$

$$= \mathop{\mathbb{P}}_{\boldsymbol{X} \sim \mathcal{P}^n, \Pi \sim \mathsf{I}(\boldsymbol{X})} ((\boldsymbol{X}, \Pi) \in E).$$

$\qquad\square$

**Lemma 3.1.** *Suppose $\mathsf{M}$ is $(\{\alpha_t\}, \beta)$-snapshot accurate for $[0, 1]$-bounded queries. Then for any $c > 0$, with probability $1 - \frac{\beta}{c}$ with respect to the randomness in the dataset $\boldsymbol{X} \sim \mathcal{P}^n$ and transcript $\Pi \sim \mathsf{I}(\boldsymbol{X}; \mathsf{A}, \mathsf{M})$, we have for all $t \in [\![n_0, n]\!]$ that $\max_{(Q,R) \in \Pi_t} |R - Q(\mathcal{Q}_\Pi^t)| \leq \alpha_t + c$.*

*Proof.* Given a transcript $\pi \in \mathcal{T}$, let a round-query-estimate tuple that achieves the largest $\alpha_t$-adjusted posterior error be denoted

$$t^\star, q^\star, r^\star \in \mathop{\arg\max}_{t \in [\![n_0, n]\!], (q,r) \in \pi_t} |r - q(\mathcal{Q}_\pi^t)| - \alpha_t,$$

where we have omitted the dependence on $\pi$. We use this definition to write the probability of interest in terms of a single event, which we then express as a union of two independent (since $\alpha_{t^\star} + c > 0$) events corresponding to the branches of the absolute value function:

$$\mathop{\mathbb{P}}_{\boldsymbol{X} \sim \mathcal{P}^n, \Pi \sim \mathsf{I}(\boldsymbol{X})} \left( \bigcup_{t=n_0}^{n} \bigcup_{(q,r) \in \Pi_t} \left\{ |r - q(\mathcal{Q}_\Pi^t)| > \alpha_t + c \right\} \right)$$

$$= \mathop{\mathbb{P}}_{\boldsymbol{X} \sim \mathcal{P}^n, \Pi \sim \mathsf{I}(\boldsymbol{X})} \left( |r^\star - q^\star(\mathcal{Q}_\Pi^{t^\star})| - \alpha_{t^\star} > c \right)$$

$$= \mathop{\mathbb{P}}_{\boldsymbol{X} \sim \mathcal{P}^n, \Pi \sim \mathsf{I}(\boldsymbol{X})} \left( r^\star - q^\star(\mathcal{Q}_\Pi^{t^\star}) - \alpha_{t^\star} > c \right) + \mathop{\mathbb{P}}_{\boldsymbol{X} \sim \mathcal{P}^n, \Pi \sim \mathsf{I}(\boldsymbol{X})} \left( q^\star(\mathcal{Q}_\Pi^{t^\star}) - r^\star - \alpha_{t^\star} > c \right). \quad (2)$$

Observe that the first term in (2) can be bounded as follows:

$$\mathbb{P}_{\boldsymbol{X}\sim\mathcal{P}^n,\Pi\sim\mathsf{I}(\boldsymbol{X})}\left(r^\star - q^\star(\mathcal{Q}_\Pi^{t^\star}) - \alpha_{t^\star} > c\right)$$

$$= \mathbb{P}_{\boldsymbol{X}\sim\mathcal{P}^n,\Pi\sim\mathsf{I}(\boldsymbol{X})}\left(\mathbb{E}_{\boldsymbol{X}'\sim\mathcal{Q}_\Pi}\left[r^\star - q^\star(\boldsymbol{X}'_{[\![t^\star]\!]}) - \alpha_{t^\star}\right] > c\right) \tag{3}$$

$$\leq \mathbb{P}_{\boldsymbol{X}\sim\mathcal{P}^n,\Pi\sim\mathsf{I}(\boldsymbol{X})}\left(\mathbb{E}_{\boldsymbol{X}'\sim\mathcal{Q}_\Pi}\left[\max\{r^\star - q^\star(\boldsymbol{X}'_{[\![t^\star]\!]}) - \alpha_{t^\star}, 0\}\right] > c\right)$$

$$\leq \frac{1}{c}\,\mathbb{E}_{\boldsymbol{X}\sim\mathcal{P}^n,\Pi\sim\mathsf{I}(\boldsymbol{X})}\left[\mathbb{E}_{\boldsymbol{X}'\sim\mathcal{Q}_\Pi}\left[\max\{r^\star - q^\star(\boldsymbol{X}'_{[\![t^\star]\!]}) - \alpha_{t^\star}, 0\}\right]\right] \tag{4}$$

$$\leq \frac{1}{c}\,\mathbb{E}_{\boldsymbol{X}\sim\mathcal{P}^n,\Pi\sim\mathsf{I}(\boldsymbol{X})}\left[\mathbb{P}_{\boldsymbol{X}'\sim\mathcal{Q}_\Pi}\left(r^\star - q^\star(\boldsymbol{X}'_{[\![t^\star]\!]}) - \alpha_{t^\star} > 0\right)\right] \tag{5}$$

$$= \frac{1}{c}\,\mathbb{P}_{\boldsymbol{X}\sim\mathcal{P}^n,\Pi\sim\mathsf{I}(\boldsymbol{X}),\boldsymbol{X}'\sim\mathcal{Q}_\Pi}\left(r^\star - q^\star(\boldsymbol{X}'_{[\![t^\star]\!]}) - \alpha_{t^\star} > 0\right)$$

$$= \frac{1}{c}\,\mathbb{P}_{\boldsymbol{X}\sim\mathcal{P}^n,\Pi\sim\mathsf{I}(\boldsymbol{X})}\left(r^\star - q^\star(\boldsymbol{X}_{[\![t^\star]\!]}) - \alpha_{t^\star} > 0\right) \tag{6}$$

where line (3) follows from the definition of $q_{t,j}(\mathcal{Q}_\Pi^t)$; line (4) follows from Markov's inequality; line (5) follows from the fact that $r - q(\boldsymbol{X}'_{[\![t]\!]}) - \alpha_t \leq 1$ for a $[0,1]$-bounded query and mechanism; and line (6) follows from Lemma B.1. By symmetry, a similar bound holds for the second term in (2).

Substituting these bounds in (2) gives

$$\mathbb{P}_{\boldsymbol{X}\sim\mathcal{P}^n,\Pi\sim\mathsf{I}(\boldsymbol{X})}\left(\bigcup_{t=n_0}^{n}\bigcup_{(q,r)\in\Pi_t}\{|r - q(\mathcal{Q}_\Pi^t)| > \alpha_t + c\}\right)$$

$$\leq \frac{1}{c}\,\mathbb{P}_{\boldsymbol{X}\sim\mathcal{P}^n,\Pi\sim\mathsf{I}(\boldsymbol{X})}\left(|r^\star - q^\star(\boldsymbol{X}_{[\![t^\star]\!]})| - \alpha_{t^\star} > 0\right) \tag{7}$$

$$\leq \frac{1}{c}\,\mathbb{P}_{\boldsymbol{X}\sim\mathcal{P}^n,\Pi\sim\mathsf{I}(\boldsymbol{X})}\left(\bigcup_{t=n_0}^{n}\bigcup_{(q,r)\in\Pi_t}\{|r - q(\boldsymbol{X}_{[\![t]\!]})| > \alpha_t\}\right) \tag{8}$$

$$\leq \frac{\beta}{c}, \tag{9}$$

where line (7) follows from the independence of the events in the two terms; line (8) follows since the starred round-query-estimate tuple may not achieve the largest $\alpha_t$-adjusted snapshot error; and line (9) follows from the definition of $(\{\alpha_t\}, \beta)$-snapshot accuracy. □

**Theorem 3.2** (PS transfer theorem). *Suppose* $\mathsf{M}$ *is an* $(\{\alpha_t\}, \beta)$-*snapshot accurate mechanism for* $[0,1]$-*bounded queries and* $\mathsf{I}(\,\cdot\,;\,\cdot\,,\mathsf{M})$ *is an* $(\epsilon, \delta)$-*posterior stable interaction. Then for every* $c > 0$, $\mathsf{M}$ *is* $(\{\alpha_t'\}, \beta')$-*distributionally accurate for* $\alpha_t' = \alpha_t + c + \epsilon$ *and* $\beta' = \frac{\beta}{c} + \delta$.

*Proof.* Given a transcript $\pi \in \mathcal{T}$, let a round-query-estimate tuple that achieves the largest $\alpha_t$-adjusted distributional error be denoted

$$t^\star, q^\star, r^\star \in \argmax_{t\in[\![n_0,n]\!],(q,r)\in\pi_t} |r - q(\mathcal{P}^t)| - \alpha_t,$$

where we have omitted the dependence on $\pi$. Using this definition, we express the probability of interest in terms of a single event, and then obtain an upper bound using the triangle inequality and

the union bound:

$$\mathop{\mathbb{P}}_{\boldsymbol{X}\sim\mathcal{P}^n,\Pi\sim\mathsf{I}(\boldsymbol{X})}\left(\bigcup_{t=n_0}^{n}\bigcup_{(q,r)\in\Pi_t}\{|r-q(\mathcal{P}^t)|>\alpha_t+c+\epsilon\}\right)$$

$$=\mathop{\mathbb{P}}_{\boldsymbol{X}\sim\mathcal{P}^n,\Pi\sim\mathsf{I}(\boldsymbol{X})}\left(|r^\star-q^\star(\mathcal{P}^{t^\star})|-\alpha_{t^\star}>c+\epsilon\right)$$

$$\leq\mathop{\mathbb{P}}_{\boldsymbol{X}\sim\mathcal{P}^n,\Pi\sim\mathsf{I}(\boldsymbol{X})}\left(|r^\star-q^\star(\mathcal{Q}_\Pi^{t^\star})|-\alpha_{t^\star}+|q^\star(\mathcal{Q}_\Pi^{t^\star})-q^\star(\mathcal{P}^{t^\star})|>c+\epsilon\right)$$

$$\leq\mathop{\mathbb{P}}_{\boldsymbol{X}\sim\mathcal{P}^n,\Pi\sim\mathsf{I}(\boldsymbol{X})}\left(|r^\star-q^\star(\mathcal{Q}_\Pi^{t^\star})|-\alpha_{t^\star}>c\right)$$

$$+\mathop{\mathbb{P}}_{\boldsymbol{X}\sim\mathcal{P}^n,\Pi\sim\mathsf{I}(\boldsymbol{X})}\left(|q^\star(\mathcal{Q}_\Pi^{t^\star})-q^\star(\mathcal{P}^{t^\star})|>\epsilon\right). \tag{10}$$

We can upper bound the two probabilities in (10) by maximizing the LHS of the inequalities with respect to $t^\star,q^\star,r^\star\in\bigcup_{t=n_0}^n\{(t,q,r):(q,r)\in\Pi_t\}$. Then Lemma 3.1 upper bounds the first probability by $\frac{\beta}{c}$ and Definition 2.3 bounds the second probability by $\delta$, giving the required result. $\qquad\square$

**Lemma 3.3.** *An $(\epsilon,\delta)$-DP interaction $\mathsf{I}(\,\cdot\,;\,\cdot\,,\mathsf{M})$ for statistical queries is $(\epsilon',\delta')$-PS for $\epsilon'=\mathrm{e}^\epsilon-1+2c\sum_{t=1}^n\delta(t)/n_0$, $\delta'=1/c$ and any $c>0$.*

*Proof.* Given a transcript $\pi\in\mathcal{T}$, let the round-query-estimate tuple that achieves the largest absolute difference be denoted

$$t^\star,q^\star,r^\star\in\mathop{\arg\max}_{t\in[\![n_0,n]\!],(q,r)\in\pi_t}|q(\mathcal{Q}_\pi^t)-q(\mathcal{P}^t)|,$$

where we have omitted the dependence on $\pi$.

Define for any $\alpha>0$, $x\in\mathcal{X}$, $t\in[\![n]\!]$, $z\in[0,1/n_0]$:

$$\boldsymbol{\Pi}(\alpha)=\left\{\pi\in\mathcal{T}:q^\star(\mathcal{Q}_\pi^{t^\star})-q^\star(\mathcal{P}^{t^\star})>\alpha\right\},$$

$$\mathcal{X}^+(\pi)=\left\{x\in\mathcal{X}:\mathop{\mathbb{P}}_{\boldsymbol{X}'\sim\mathcal{Q}_\pi,t\sim\mathcal{U}_{[\![n_0,t^\star]\!]}}(X'_\tau=x)\geq\mathop{\mathbb{P}}_{X\sim\mathcal{P}}(X=x)\right\},$$

$$B^+(\alpha)=\bigcup_{\pi\in\boldsymbol{\Pi}(\alpha)}(\mathcal{X}^+(\pi)\times\{\pi\}),$$

$$\boldsymbol{\Pi}^+(\alpha,x)=\{\pi\in\mathcal{T}:(x,\pi)\in B^+(\alpha)\},$$

$$\boldsymbol{\Pi}^+(\alpha,x,z,t)=\left\{\pi\in\boldsymbol{\Pi}^+(\alpha,x):t\leq t^\star,1>zt^\star\right\}.$$

Fix any $\alpha>0$ and let $\tilde{\delta}=\sum_{t=1}^n\delta(t)$. Suppose $\mathbb{P}_{\boldsymbol{X}\sim\mathcal{P}^n,\Pi\sim\mathsf{I}(\boldsymbol{X})}\left(|q^\star(\mathcal{Q}_\Pi^{t^\star})-q^\star(\mathcal{P}^{t^\star})|>\alpha\right)>\frac{1}{c}$, which implies either $\mathbb{P}_{\boldsymbol{X}\sim\mathcal{P}^n,\Pi\sim\mathsf{I}(\boldsymbol{X})}\left(q^\star(\mathcal{Q}_\Pi^{t^\star})-q^\star(\mathcal{P}^{t^\star})>\alpha\right)>\frac{1}{2c}$ or $\mathbb{P}_{\boldsymbol{X}\sim\mathcal{P}^n,\Pi\sim\mathsf{I}(\boldsymbol{X})}\left(q^\star(\mathcal{P}^{t^\star})-q^\star(\mathcal{Q}_\Pi^{t^\star})>\alpha\right)>\frac{1}{2c}$. Without loss of generality assume

$$\mathop{\mathbb{P}}_{\boldsymbol{X}\sim\mathcal{P}^n,\Pi\sim\mathsf{I}(\boldsymbol{X})}\left(q^\star(\mathcal{Q}_\Pi^{t^\star})-q^\star(\mathcal{P})>\alpha\right)=\mathbb{P}(\Pi\in\boldsymbol{\Pi}(\alpha))>\frac{1}{2c}. \tag{11}$$

From the definition of $\boldsymbol{\Pi}(\alpha)$, we have

$$\alpha\,\mathbb{P}(\Pi\in\boldsymbol{\Pi}(\alpha))<\sum_{\pi\in\boldsymbol{\Pi}(\alpha)}\{q^\star(\mathcal{Q}_\pi^{t^\star})-q^\star(\mathcal{P}^{t^\star})\}$$

$$=\sum_{\pi\in\boldsymbol{\Pi}(\alpha)}\left\{\mathop{\mathbb{E}}_{\boldsymbol{X}'\sim\mathcal{Q}_\pi}[q^\star(\boldsymbol{X}'_{[\![t^\star]\!]})]-\mathop{\mathbb{E}}_{\boldsymbol{X}\sim\mathcal{P}^n}[q^\star(\boldsymbol{X}_{[\![t^\star]\!]})]\right\}$$

$$=\sum_{\pi\in\boldsymbol{\Pi}(\alpha)}\left\{\mathop{\mathbb{E}}_{\boldsymbol{X}'\sim\mathcal{Q}_\pi,t\sim\mathcal{U}_{[\![t^\star]\!]}}[q^\star(X'_t)]-\mathop{\mathbb{E}}_{X\sim\mathcal{P}}[q^\star(X)]\right\}, \tag{12}$$

where we have used linearity of statistical queries in the last line. Expanding out the difference of expectations, we have

$$\mathop{\mathbb{E}}_{\boldsymbol{X}'\sim\mathcal{Q}_\pi,t\sim\mathcal{U}_{[\![t^\star]\!]}}[q^\star(X_t')] - \mathop{\mathbb{E}}_{X\sim\mathcal{P}}[q^\star(X)]$$

$$= \sum_{x\in\mathcal{X}} q^\star(x)\left\{\frac{1}{t^\star}\sum_{t=1}^{t^\star}\mathbb{P}(X_t' = x \mid \Pi = \pi) - \mathbb{P}(X = x)\right\}$$

$$\leq \sum_{x\in\mathcal{X}^+(\pi)}\left\{\frac{1}{t^\star}\sum_{t=1}^{t^\star}\mathbb{P}(X_t' = x \mid \Pi = \pi) - \mathbb{P}(X = x)\right\} \tag{13}$$

$$= \sum_{x\in\mathcal{X}^+(\pi)}\frac{\mathbb{P}(X = x)}{\mathbb{P}(\Pi = \pi)}\left\{\frac{1}{t^\star}\sum_{t=1}^{t^\star}\mathbb{P}(\Pi = \pi \mid X_t' = x) - \mathbb{P}(\Pi = \pi)\right\} \tag{14}$$

where (13) follows by dropping negative terms from the sum and using the boundedness of statistical queries, and (14) follows from Bayes' theorem.

Putting (14) in (12) and swapping the order of the sums gives

$$\alpha\,\mathbb{P}(\Pi \in \boldsymbol{\Pi}(\alpha))$$

$$< \sum_{\pi\in\boldsymbol{\Pi}(\alpha)}\sum_{x\in\mathcal{X}^+(\pi)}\mathbb{P}(X = x)\left\{\frac{1}{t^\star}\sum_{t=1}^{t^\star}\mathbb{P}(\Pi = \pi \mid X_t' = x) - \mathbb{P}(\Pi = \pi)\right\}$$

$$= \sum_{t=1}^n\sum_{x\in\mathcal{X}}\mathbb{P}(X = x)\sum_{\pi\in\boldsymbol{\Pi}^+(\alpha,x)}\frac{1}{t^\star}\mathbf{1}_{[t\leq t^\star]}\left\{\mathbb{P}(\Pi = \pi \mid X_t' = x) - \mathbb{P}(\Pi = \pi)\right\}$$

$$= \sum_{t=1}^n\sum_{x\in\mathcal{X}}\mathbb{P}(X = x)\sum_{\pi\in\boldsymbol{\Pi}^+(\alpha,x)}\int_0^{\frac{1}{n_0}}\mathbf{1}_{\left[\frac{1}{t^\star}>z\right]}\,dz\,\mathbf{1}_{[t\leq t^\star]}\left\{\mathbb{P}(\Pi = \pi \mid X_t' = x) - \mathbb{P}(\Pi = \pi)\right\}$$

$$= \sum_{t=1}^n\sum_{x\in\mathcal{X}}\mathbb{P}(X = x)\int_0^{\frac{1}{n_0}}\left\{\mathbb{P}(\Pi \in \boldsymbol{\Pi}^+(\alpha,x,z,t) \mid X_t' = x) - \mathbb{P}(\Pi \in \boldsymbol{\Pi}^+(\alpha,x,z,t))\right\}\,dz$$

$$\leq \sum_{t=1}^n\sum_{x\in\mathcal{X}}\mathbb{P}(X = x)\int_0^{\frac{1}{n_0}}\left\{(\mathrm{e}^\epsilon - 1)\,\mathbb{P}(\Pi \in \boldsymbol{\Pi}^+(\alpha,x,z,t)) + \boldsymbol{\delta}(t)\right\}\,dz \tag{15}$$

$$= \sum_{x\in\mathcal{X}}\mathbb{P}(X = x)\left\{(\mathrm{e}^\epsilon - 1)\,\mathbb{P}(\Pi \in \boldsymbol{\Pi}^+(\alpha,x)) + \frac{\tilde{\delta}}{n_0}\right\}$$

$$= (\mathrm{e}^\epsilon - 1)\,\mathbb{P}(\Pi \in \boldsymbol{\Pi}(\alpha)) + \frac{\tilde{\delta}}{n_0}$$

$$< \left(\mathrm{e}^\epsilon - 1 + \frac{2c\tilde{\delta}}{n_0}\right)\mathbb{P}(\Pi \in \boldsymbol{\Pi}(\alpha)) \tag{16}$$

where (15) follows from Lemma B.2, and (16) follows from (11). This is a contradiction for $\alpha \geq \mathrm{e}^\epsilon - 1 + 2c\tilde{\delta}/n_0$. $\qquad\square$

**Lemma 3.5.** *An $(\epsilon, \boldsymbol{\delta})$-DP interaction $\mathsf{I}(\,\cdot\,;\,\cdot\,,\mathsf{M})$ for $\boldsymbol{\Delta}$-sensitive queries is $(\epsilon', \delta')$-posterior stable for $\epsilon' = \mathrm{e}^\epsilon\max_{\tau_1\in[\![n_0,n]\!]}\tau_1\Delta_{\tau_1} - \min_{\tau_2\in[\![n_0,n]\!]}\tau_2\Delta_{\tau_2} + 4c\left(\sum_{t=1}^n\boldsymbol{\delta}(t)\right)\max_{\tau_3\in[\![n_0,n]\!]}\Delta_{\tau_3}$, $\delta' = 1/c$ and any $c > 0$.*

*Proof.* Given a transcript $\pi \in \mathcal{T}$, let a round-query-result tuple that achieves the largest absolute difference be denoted

$$t^\star, q^\star, r^\star \in \mathop{\arg\max}_{t\in[\![n_0,n]\!],(q,r)\in\pi_t}|q(\mathcal{Q}_\pi^t) - q(\mathcal{P}^t)|,$$

where the dependence on $\pi$ is omitted.

Let $+$ denote the concatenation operator, such that for any pair of sequences $\boldsymbol{x}$ and $\boldsymbol{y}$ of length $n$ and $m$ respectively, we have $\boldsymbol{x}+\boldsymbol{y} := (x_1, \ldots, x_n, y_1, \ldots, y_m)$. Let $\Delta^\star = \max_{t \in [\![n_0, n]\!]} \Delta_t$ and $\tilde{\delta} = \sum_{t=1}^n \boldsymbol{\delta}(t)$. For any $\alpha \geq 0$, $\tau, t \in [\![n]\!]$, $\boldsymbol{x} \in \mathcal{X}^n$ and $z \in [0, 2\Delta^\star]$, define

$$v(q, \tau, \boldsymbol{x}_{[\![t]\!]}) := \mathbb{E}_{\boldsymbol{X}' \sim \mathcal{P}^{\max\{\tau - t, 0\}}} \left[ q(\boldsymbol{x}_{[\![t]\!]} + \boldsymbol{X}') \right]$$

$$\boldsymbol{\Pi}(\alpha) := \left\{ \pi \in \mathcal{T} : q^\star(\mathcal{Q}_\pi^{t^\star}) - q^\star(\mathcal{P}^{t^\star}) > \alpha \right\},$$

$$\boldsymbol{\Pi}(\alpha, z, \boldsymbol{x}_{[\![t]\!]}) := \left\{ \pi \in \boldsymbol{\Pi}(\alpha) : \mathbf{1}_{[t \leq t^\star]} \left( v(q^\star, t^\star, \boldsymbol{x}_{[\![t]\!]}) - v(q^\star, t^\star, \boldsymbol{x}_{[\![t-1]\!]}) + \Delta_{t^\star} \right) > z \right\}.$$

Using the definition of differential privacy, observe that

$$\sum_{\pi \in \boldsymbol{\Pi}(\alpha)} \mathbb{P}(\Pi = \pi \mid \boldsymbol{X} = \boldsymbol{x}) \mathbf{1}_{[t \leq t^\star]} \left( v(q^\star, t^\star, \boldsymbol{x}_{[\![t]\!]}) - v(q^\star, t^\star, \boldsymbol{x}_{[\![t-1]\!]}) + \Delta_{t^\star} \right)$$

$$= \sum_{\pi \in \boldsymbol{\Pi}(\alpha)} \mathbb{P}(\Pi = \pi \mid \boldsymbol{X} = \boldsymbol{x}) \int_0^{2\Delta^\star} \mathbf{1}_{\left[ \mathbf{1}_{[t \leq t^\star]} (v(q^\star, t^\star, \boldsymbol{x}_{[\![t]\!]}) - v(q^\star, t^\star, \boldsymbol{x}_{[\![t-1]\!]}) + \Delta_{t^\star}) > z \right]} \, dz$$

$$= \int_0^{2\Delta^\star} \mathbb{P}(\Pi \in \boldsymbol{\Pi}(\alpha, z, \boldsymbol{x}_{[\![t]\!]}) \mid \boldsymbol{X} = \boldsymbol{x}) \, dz$$

$$\leq \int_0^{2\Delta^\star} \left( e^\epsilon \, \mathbb{P}(\Pi \in \boldsymbol{\Pi}(\alpha, z, \boldsymbol{x}_{[\![t]\!]}) \mid \boldsymbol{X} = \mathrm{rep}(\boldsymbol{x}, t, x')) + \boldsymbol{\delta}(t) \right) \, dz$$

$$= e^\epsilon \sum_{\pi \in \boldsymbol{\Pi}(\alpha)} \mathbb{P}(\Pi = \pi \mid \boldsymbol{X} = \mathrm{rep}(\boldsymbol{x}, t, x')) \mathbf{1}_{[t \leq t^\star]} \left( v(q^\star, t^\star, \boldsymbol{x}_{[\![t]\!]}) - v(q^\star, t^\star, \boldsymbol{x}_{[\![t-1]\!]}) + \Delta_{t^\star} \right)$$

$$+ 2\Delta^\star \boldsymbol{\delta}(t) \tag{17}$$

where $\mathrm{rep}(\boldsymbol{x}, t, x')$ denotes the dataset obtained from $\boldsymbol{x}$ by replacing the $t$-th data point with $x' \in \mathcal{X}$. Taking the expectation of inequality (17) with respect to $\boldsymbol{X} \sim \mathcal{P}^n$ and $X' \sim \mathcal{P}$, and summing over $t \in [\![n]\!]$, we have

$$\sum_{t=1}^n \mathbb{E}_{\boldsymbol{X} \sim \mathcal{P}^n} \left[ \sum_{\pi \in \boldsymbol{\Pi}(\alpha)} \mathbb{P}(\Pi = \pi \mid \boldsymbol{X}) \mathbf{1}_{[t \leq t^\star]} \left( v(q^\star, t^\star, \boldsymbol{X}_{[\![t]\!]}) - v(q^\star, t^\star, \boldsymbol{X}_{[\![t-1]\!]}) + \Delta_{t^\star} \right) \right]$$

$$\leq \sum_{t=1}^n \mathbb{E}_{\boldsymbol{X} \sim \mathcal{P}^n, X \sim \mathcal{P}} \left[ e^\epsilon \sum_{\pi \in \boldsymbol{\Pi}(\alpha)} \mathbb{P}(\Pi = \pi \mid \boldsymbol{X}) \right.$$

$$\left. \times \mathbf{1}_{[t \leq t^\star]} \left( v(q^\star, t^\star, \mathrm{rep}(\boldsymbol{x}, t, x')) - v(q^\star, t^\star, \boldsymbol{x}_{[\![t-1]\!]}) + \Delta_{t^\star} \right) + 2\Delta^\star \delta \right] \tag{18}$$

$$\leq \sum_{t=1}^n \mathbb{E}_{\boldsymbol{X} \sim \mathcal{P}^n} \left[ e^\epsilon \sum_{\pi \in \boldsymbol{\Pi}(\alpha)} \mathbb{P}(\Pi = \pi \mid \boldsymbol{X}) \mathbf{1}_{[t \leq t^\star]} \Delta_{t^\star} + 2\Delta^\star \boldsymbol{\delta}(t) \right] \tag{19}$$

$$\leq e^\epsilon \max_{t \in [\![n_0, n]\!]} t\Delta_t \, \mathbb{P}(\Pi \in \boldsymbol{\Pi}(\alpha)) + 2\Delta^\star \tilde{\delta} \tag{20}$$

where (18) follows since $(\boldsymbol{X}, X')$ and $(\mathrm{rep}(\boldsymbol{X}, t, X'), X_t)$ have the same distribution, and (19) follows since $X'$ is independent of $\Pi$ and $\mathbb{E}_{X \sim \mathcal{P}}[v(q^\star, t^\star, \mathrm{rep}(\boldsymbol{X}, t, X))] = v(q^\star, t^\star, \boldsymbol{X}_{[\![t-1]\!]})$.

Subtracting $\sum_{\pi \in \boldsymbol{\Pi}(\alpha)} t^\star \Delta_{t^\star}$ from both sides of inequality (20), we have

$$\sum_{t=1}^n \mathbb{E}_{\boldsymbol{X} \sim \mathcal{P}^n} \left[ \sum_{\pi \in \boldsymbol{\Pi}(\alpha)} \mathbb{P}(\Pi = \pi \mid \boldsymbol{X}) \mathbf{1}_{[t \leq t^\star]} \left( v(q^\star, t^\star, \boldsymbol{X}_{[\![t]\!]}) - v(q^\star, t^\star, \boldsymbol{X}_{[\![t-1]\!]}) \right) \right]$$

$$\leq \left( e^\epsilon \max_{t \in [\![n_0, n]\!]} t\Delta_t - \min_{t \in [\![n_0, n]\!]} t\Delta_t \right) \mathbb{P}(\Pi \in \boldsymbol{\Pi}(\alpha)) + 2\Delta^\star \tilde{\delta}.$$

Now fix $\alpha = e^\epsilon \max_{t \in [\![n_0, n]\!]} t\Delta_t - \min_{t' \in [\![n_0, n]\!]} t'\Delta_{t'} + 4cn \max_{t'' \in [\![n_0, n]\!]} \Delta_{t''}$. Suppose $\mathbb{P}(|q^\star(\mathcal{Q}_\Pi^{t^\star}) - q^\star(\mathcal{P}^{t^\star})| > \alpha) > \frac{1}{c}$. Then it must be that either $\mathbb{P}(q^\star(\mathcal{Q}_\Pi^{t^\star}) - q^\star(\mathcal{P}^{t^\star}) > \alpha) > \frac{1}{2c}$ or

$\mathbb{P}(q^\star(\mathcal{P}^{t^\star}) - q^\star(\mathcal{Q}_\Pi^{t^\star}) > \alpha) > \frac{1}{2c}$. Without loss of generality, assume

$$\mathbb{P}(q^\star(\mathcal{Q}_\Pi^{t^\star}) - q^\star(\mathcal{P}^{t^\star}) > \alpha) = \mathbb{P}(\Pi \in \mathbf{\Pi}(\alpha)) > \frac{1}{2c}.$$

But this leads to a contradiction since

$$\alpha\,\mathbb{P}(\Pi \in \mathbf{\Pi}(\alpha))$$
$$< \sum_{\pi \in \mathbf{\Pi}(\alpha)} \mathbb{P}(\Pi = \pi)\left(q^\star(\mathcal{Q}_\pi^{t^\star}) - q^\star(\mathcal{P}^{t^\star})\right)$$
$$= \mathop{\mathbb{E}}_{\mathbf{X}\sim\mathcal{P}^n}\left[\sum_{\pi \in \mathbf{\Pi}(\alpha)} \mathbb{P}(\Pi = \pi \mid \mathbf{X})\left(q^\star(\mathbf{X}_{[\![t^\star]\!]}) - q^\star(\mathcal{P}^{t^\star})\right)\right]$$
$$= \sum_{t=1}^{n} \mathop{\mathbb{E}}_{\mathbf{X}\sim\mathcal{P}^n}\left[\sum_{\pi \in \mathbf{\Pi}(\alpha)} \mathbb{P}(\Pi = \pi \mid \mathbf{X})\mathbf{1}_{[t \le t^\star]}\left(v(q^\star, t^\star, \mathbf{X}_{[\![t]\!]}) - v(q^\star, t^\star, \mathbf{X}_{[\![t-1]\!]})\right)\right]$$
$$\le \left(\mathsf{e}^\epsilon \max_{t\in[\![n_0,n]\!]} t\Delta_t - \min_{t'\in[\![n_0,n]\!]} t'\Delta_{t'}\right)\mathbb{P}(\Pi \in \mathbf{\Pi}(\alpha)) + 2\Delta^\star\tilde{\delta}$$
$$\le \mathbb{P}(\Pi \in \mathbf{\Pi}(\alpha))\left(\mathsf{e}^\epsilon \max_{t\in[\![n_0,n]\!]} t\Delta_t - \min_{t'\in[\![n_0,n]\!]} t'\Delta_{t'} + 4c\Delta^\star\tilde{\delta}\right)$$

$\square$

**Lemma B.2** (Lemma 21, [15]). *If* $\mathsf{I}(\,\cdot\,;\,\cdot\,,\mathsf{M})$ *is* $(\epsilon, \boldsymbol{\delta})$-*differentially private, then for any event* $E \in \mathcal{T}$, *any index* $t \in [\![n]\!]$ *and value* $x \in \mathcal{X}$:

$$\mathop{\mathbb{P}}_{\mathbf{X}\sim\mathcal{P}^n, \Pi\sim\mathsf{I}(\mathrm{rep}(\mathbf{X},t,x))}[\Pi \in E] \le \mathsf{e}^\epsilon \mathop{\mathbb{P}}_{\mathbf{X}\sim\mathcal{P}^n, \Pi\sim\mathsf{I}(\mathbf{X})}[\Pi \in E] + \boldsymbol{\delta}(t)$$

*where* $\mathrm{rep}(\mathbf{X}, t, x)$ *is the dataset obtained from* $\mathbf{X}$ *by replacing the* $t$-*th data point with* $x$.

*Proof.* This follows from expanding the definitions

$$\mathop{\mathbb{P}}_{\mathbf{X}\sim\mathcal{P}^n, \Pi\sim\mathsf{I}(\mathrm{rep}(\mathbf{X},t,x))}[\Pi \in E] = \sum_{\mathbf{x}\in\mathcal{X}^n} \mathop{\mathbb{P}}_{\mathbf{X}\sim\mathcal{P}^n}[\mathbf{X} = \mathbf{x}] \mathop{\mathbb{P}}_{\Pi\sim\mathsf{I}(\mathrm{rep}(\mathbf{x},t,x))}[\Pi \in E]$$
$$\le \sum_{\mathbf{x}\in\mathcal{X}^n} \mathop{\mathbb{P}}_{\mathbf{X}\sim\mathcal{P}^n}[\mathbf{X} = \mathbf{x}]\left(\mathsf{e}^\epsilon \mathop{\mathbb{P}}_{\Pi\sim\mathsf{I}(\mathbf{x})}[\Pi \in E] + \boldsymbol{\delta}(t)\right) \qquad (21)$$
$$= \mathsf{e}^\epsilon \mathop{\mathbb{P}}_{\mathbf{X}\sim\mathcal{P}^n, \Pi\sim\mathsf{I}(\mathbf{X})}[\Pi \in E] + \boldsymbol{\delta}(t)$$

where (21) follows from the definition of differential privacy. $\square$

## C  Proofs for Section 4

### C.1  Generalization Guarantees Assuming Fixed Privacy Parameters

**Lemma 4.2.** *Consider an interaction* $\mathsf{I}(\cdot;\cdot,\mathsf{M})$ *where* $\mathsf{M}$ *is the ordinary or clipped Gaussian mechanism. Suppose the analyst decides to submit* $k_\tau$ *statistical queries in round* $\tau \in [\![n_0, n]\!]$ *before* $\mathsf{I}(\cdot;\cdot,\mathsf{M})$ *is executed. Then* $\mathsf{I}(\cdot;\cdot,\mathsf{M})$ *satisfies* $(\epsilon,\boldsymbol{\delta})$-*DP for any* $\epsilon > 0$ *and* $\boldsymbol{\delta}(t) \le \psi(\gamma^\star, \boldsymbol{\rho}(t), \epsilon)$, *where* $\boldsymbol{\rho}(t) = \sum_{\tau=n_0}^{n} k_\tau \mathbf{1}_{[t\le\tau]}/2\sigma_\tau^2\tau^2$, $\psi(\gamma,\rho,\epsilon) = \mathsf{e}^{(\gamma-1)(\gamma\rho-\epsilon)}\left(1 - \gamma^{-1}\right)^\gamma/(\gamma - 1)$ *and* $\gamma^\star = \arg\min_{\gamma\in(1,\infty)} \psi(\gamma, \max_{t\in[\![n]\!]} \boldsymbol{\rho}(t), \epsilon)$.

*Proof.* We begin by bounding the privacy of the interaction using zero-concentrated differential privacy (zCDP), as defined in Definition D.3. When releasing an estimate to a single statistical query in round $\tau$, the Gaussian mechanism satisfies $\rho$-zCDP with $\rho(t) = \mathbf{1}_{[t\le\tau]}/2\sigma_\tau^2\tau^2$, where we have used $1/\tau^2$ as an upper bound on the sensitivity of the query [37, Proposition 1.6]. By post-processing (see Theorem D.5), clipping the output of the Gaussian mechanism does not accrue an additional privacy loss, nor does the analyst's choice of the next query conditioned on the previous

releases. Now by advanced composition (see Theorem D.4), the interaction satisfies $\rho$-zCDP with $\rho(t) = \sum_{\tau=n_0}^{n} k_\tau \mathbf{1}_{[t \leq \tau]}/2\sigma_\tau^2 \tau^2$. We note that in order to apply this theorem, we have relied on the fact that the privacy parameters are non-adaptive, which is true when the $k_t$'s are fixed in advance. Finally we convert from $\rho$-zCDP to $(\epsilon, \delta)$-DP using Corollary D.9. $\qquad \square$

**Lemma 4.3.** *For any* $\beta \in (0, 1)$, *the clipped Gaussian mechanism with* $\sigma_t \propto \alpha_t$ *is* $(\{\alpha_t\}, \beta)$-*snapshot accurate for* $k$ *queries with*

$$\frac{\alpha_t}{\sqrt{2}\sigma_t} = \mathrm{erfc}^{-1}\left(2 - 2\left(1 - \frac{\beta}{2}\right)^{\frac{1}{k}}\right) < \mathrm{erfc}^{-1}\left(\frac{\beta}{k}\right).$$

*Proof.* Let $k_t$ denote the number of queries asked in round $t$. Let $j \in \{1, \ldots, k_t\}$ index the queries asked in round $t$ and $Z_{t,j} \overset{\text{iid.}}{\sim} \mathrm{Normal}(0, \sigma_t)$ denote the Gaussian noise added by the mechanism to the $j$-th query in round $t$.

We begin with the observation that

$$\mathbb{P}\left(\max_{t \in [\![n_0, n]\!]} \max_{j \in [\![k_t]\!]} (-Z_{t,j} - \alpha_t) \geq 0\right) = 1 - \mathbb{P}\left(\max_{t \in [\![n_0, n]\!]} \max_{j \in [\![k_t]\!]} (-Z_{t,j} - \alpha_t) < 0\right)$$

$$= 1 - \prod_{t=n_0}^{n} \prod_{j=1}^{k_t} \mathbb{P}(Z_{t,j} \geq -\alpha_t)$$

$$= 1 - \prod_{t=n_0}^{n} \prod_{j=1}^{k_t} \left(1 - \frac{1}{2}\mathrm{erfc}\left(\frac{\alpha_t}{\sqrt{2}\sigma_t}\right)\right)$$

$$= 1 - \left(1 - \frac{1}{2}\mathrm{erfc}\left(\frac{\alpha_t}{\sqrt{2}\sigma_t}\right)\right)^k, \tag{22}$$

where the last equality follows from the fact that $\alpha_t/\sigma_t$ is constant with respect to $t$. By symmetry, this equality also holds under the replacement $Z_{t,j} \to -Z_{t,j}$.

Conditioning on the number of queries $k_t$ asked in each round $t$, we have with respect to the joint distribution on the dataset $\boldsymbol{X}$ and transcript $\Pi$ that

$$\mathbb{P}\left(\bigcup_{t=n_0}^{n} \bigcup_{(q,r) \in \Pi_t} \{|q(\boldsymbol{X}_{[\![t]\!]}) - r| \geq \alpha_t\} \,\middle|\, \bigcap_{t=n_0}^{n} \{|\Pi_t| = k_t\}\right)$$

$$= \mathbb{P}\left(\max_{t \in [\![n_0, n]\!]} \max_{j \in [\![k_t]\!]} \left\{|q_{t,j}(\boldsymbol{X}_{[\![t]\!]}) - \mathrm{clip}_{[0,1]}(q_{t,j}(\boldsymbol{X}_{[\![t]\!]}) + Z_{t,j})| - \alpha_t\right\} \geq 0\right)$$

$$\leq \mathbb{P}\left(\max_{t \in [\![n_0, n]\!]} \max_{j \in [\![k_t]\!]} \{|Z_{t,j}| - \alpha_t\} \geq 0\right)$$

$$\leq \mathbb{P}\left(\max_{t \in [\![n_0, n]\!]} \max_{j \in [\![k_t]\!]} \{-Z_{t,j} - \alpha_t\} \geq 0\right) + \mathbb{P}\left(\max_{t \in [\![n_0, n]\!]} \max_{j \in [\![k_t]\!]} \{Z_{t,j} - \alpha_t\} \geq 0\right)$$

$$= 2\left(1 - \left(1 - \frac{1}{2}\mathrm{erfc}\left(\frac{\alpha_t}{\sqrt{2}\sigma_t}\right)\right)^k\right), \tag{23}$$

where the last line follows from (22). Note that the bound only depends on the total number of queries $k$, not the number of queries asked at each time step $k_t$, thus it serves as a bound on the probability conditioned on the event $\sum_{t=n_0}^{n} |\Pi_t| = k$. The result follows by setting (23) equal to $\beta$ and solving for $\alpha_t/\sqrt{2}\sigma_t$. $\qquad \square$

**Theorem 4.4.** *Suppose the conditions of Lemma 4.2 hold and assume* $\sigma_t = \sigma > 0$. *Then the clipped Gaussian mechanism is* $(\alpha', \beta')$-*distributionally accurate for* $k$ *statistical queries for any* $\beta' \in (0, 1)$ *and* $\alpha' = \min_{\sigma, \beta, \epsilon \in \Theta} \lambda(\sigma, \beta, \epsilon)$, *where*

$$\lambda(\sigma, \beta, \epsilon) = \sqrt{2}\sigma\,\mathrm{erfc}^{-1}\left(\frac{\beta}{k}\right) + e^\epsilon - 1 + \frac{\beta}{\beta'} + \frac{2\sum_{\tau=1}^{n} \boldsymbol{\delta}(\tau)}{n_0 \beta'} + \frac{2}{\beta'}\sqrt{\frac{2\beta \sum_{\tau=1}^{n} \boldsymbol{\delta}(\tau)}{n_0}},$$

$\Theta = \{(\sigma, \beta, \epsilon) \in \mathbb{R}^3 : \sigma > 0, 0 < \beta < 1, \epsilon \geq 0\}$ *and* $\boldsymbol{\delta}(\cdot)$ *is defined in Lemma 4.2.*

*Proof.* Combining Lemmas 4.2 and 4.3 and Theorem 3.4 implies the clipped Gaussian mechanism is $(\alpha', \beta')$-distributionally accurate for $\beta' = \beta/d + 1/c$ and

$$\alpha' = \sqrt{2}\sigma \, \mathrm{erfc}^{-1}\left(\frac{\beta}{k}\right) + \mathrm{e}^{\epsilon} - 1 + \frac{2c\sum_{\tau=1}^{n}\boldsymbol{\delta}(\tau)}{n_0} + d$$

for any $c, d > 0$ and $0 < \beta < 1$. We select the free parameters to minimize $\alpha'$. Eliminating $c$ using the constraint $\beta' = \beta/d + 1/c$ and minimizing with respect to $d$ analytically gives:

$$\alpha' = \sqrt{2}\sigma \, \mathrm{erfc}^{-1}\left(\frac{\beta}{k}\right) + \mathrm{e}^{\epsilon} - 1 + \frac{2\sum_{\tau=1}^{n}\boldsymbol{\delta}(\tau)}{n_0\beta'} + \frac{\beta}{\beta'} + \frac{2}{\beta'}\sqrt{\frac{2\beta\sum_{\tau=1}^{n}\boldsymbol{\delta}(\tau)}{n_0}}$$

Minimizing this expression with respect to the remaining free parameters gives the required result. □

### C.2 Generalization Guarantees Assuming Adaptive Privacy Parameters

In this appendix, we instantiate generalization guarantees for the clipped Gaussian mechanism where we allow the analyst to *adaptively* select how many queries to ask in each round. This is in contrast with the results of Section 4.1, where we assume the number of queries asked in each round is *fixed* before interacting with the data. To support a fully adaptive analyst, we rely on a *privacy filter*, which provides differential privacy guarantees when both the mechanisms and privacy parameters are selected adaptively. Generally speaking, a privacy filter is an algorithm that continually monitors the privacy parameters of mechanisms as they are composed. At any point, it can force the composition to terminate to ensure it satisfies a pre-specified level of privacy.

We use a privacy filter proposed by Whitehouse et al. [32] for approximate zero-concentrated differential privacy (zCDP). It achieves the same rates as advanced composition [37] assuming the privacy level is specified upfront. The filter does not support non-uniform privacy accounting—where the privacy loss is estimated separately for each data point /individual. We therefore resort to uniform accounting, which results in looser generalization bounds. We expect the filter could be adapted to support non-uniform privacy accounting, in a similar way to the Rényi filter proposed by Feldman and Zrnic [36].

To incorporate Whitehouse et al.'s privacy filter for an adaptive analysis using the clipped Gaussian mechanism, we require additional monitoring as shown in Algorithm 2. Lines 1 and 4 track the running privacy level $\bar{\rho}$ using the same additive rule as advanced composition for zCDP. If releasing a response to a query would not exceed the target privacy level (line 5), then a response to the query is released (line 7) and the analysis continues, otherwise the mechanism is terminated (line 9).

---

**Algorithm 2** Composition of Clipped Gaussian Mechanisms with zCDP Privacy Filter

**Input:** target privacy level $\rho$
1: $\bar{\rho} \leftarrow 0$
2: **for** Round $t \in [\![n_0, n]\!]$ **do**
3:     **while** Analyst has another query $q$ **do**
4:         $\bar{\rho} \leftarrow \bar{\rho} + \frac{1}{2\sigma_t^2 t^2}$
5:         **if** $\bar{\rho} \leq \rho$ **then**
6:             Sample noise: $z \sim \mathcal{N}(0, \sigma_t^2)$
7:             Return response to query: $r \leftarrow \mathrm{clip}_{[0,1]}(q(\boldsymbol{x}_{[\![t]\!]}) + z)$
8:         **else**
9:             TERMINATE
10:         **end if**
11:     **end while**
12: **end for**

---

**Lemma C.1.** *Consider an interaction* $\mathsf{I}(\cdot; \cdot, \mathsf{M})$ *where* $\mathsf{M}$ *is a composition of clipped Gaussian mechanisms tracked by a privacy filter that satisfies* $\rho$*-zCDP, as described in Algorithm 2. Then the interaction satisfies* $\rho$*-zCDP, with the possibility that it may terminate early. It also satisfies* $(\epsilon, \delta)$*-DP for any* $\epsilon \geq 0$ *with* $\delta \leq \inf_{\gamma \in (1,\infty)} \mathrm{e}^{(\gamma-1)(\gamma\rho-\epsilon)}(1 - \gamma^{-1})^{\gamma}/(\gamma - 1)$.

*Proof.* We recall that the Gaussian mechanism satisfies $1/(2\sigma_t^2 t)$-zCDP for a statistical query with sensitivity $\Delta \leq 1/t$ [37, Proposition 1.6]. By post-processing for zCDP, clipping the result of the Gaussian mechanism and selecting a query as a function of the result (and past results) does not accrue an additional cost to privacy. Hence, by fully adaptive composition for the zCDP privacy filter [32, Theorem 1], the interaction satisfies $\rho$-zCDP. Finally, we convert from $\rho$-zCDP to $(\epsilon, \delta)$-DP to obtain the final result [38, Corollary 13]. $\qquad\square$

Combining Lemma C.1 with Lemma 4.3 and Theorem 3.4 yields the following generalization guarantee. We note that this essentially matches the guarantee for the more constrained setting where the number of queries in each round is fixed upfront (Theorem 4.4).

**Theorem C.2.** *Consider an interaction* $\mathsf{I}(\cdot; \cdot, \mathsf{M})$ *where* $\mathsf{M}$ *is a composition of clipped Gaussian mechanisms tracked by a zCDP privacy filter that satisfies* $(\epsilon, \delta)$*-DP. Suppose* $\mathsf{M}$ *answers* $k$ *statistical queries without terminating and assume* $\sigma_t = \sigma > 0$. *Then* $\mathsf{M}$ *is* $(\alpha', \beta')$*-distributionally accurate for* $k$ *statistical queries with*

$$\alpha' = \min_{(\sigma,\beta,\epsilon)\in\Theta} \sqrt{2}\sigma \operatorname{erfc}^{-1}\left(\frac{\beta}{k}\right) + \mathsf{e}^\epsilon - 1 + \frac{2n\delta}{n_0\beta'} + \frac{\beta}{\beta'} + \frac{2}{\beta'}\sqrt{\frac{2\beta n\delta}{n_0}}$$

*where* $\Theta$ *is defined in Theorem 4.4 and* $\delta$ *is defined in Lemma C.1.*

*Proof.* The proof is similar to the proof of Theorem 4.4. $\qquad\square$

### C.3 Generalization Guarantees using the Static Bound of Jung et al. [15]

In this appendix, we derive a generalization guarantee for the setting considered in Section 4.2 by composing a bound for static data due to Jung et al. [15]. In doing so, we will obtain the bound that is labeled **JLNRSS** in the empirical results of Section 4.2. For completeness, we begin by restating Jung et al.'s result below, with some minor changes in notation. In particular, we make the dependence of the bound on the dataset size $n$ and number of queries $k$ explicit, since we will apply the bound to multiple batches, each with a different value of $n$ and $k$.

**Theorem C.3.** *Consider ADA on a static dataset of size* $n$—*i.e., an instantiation of Algorithm 1 with* $n_0 = n - 1$. *The clipped Gaussian mechanism can be used to answer* $k$ *statistical queries while satisfying* $(\alpha(n, k, \beta), \beta)$*-distributional accuracy for any* $\beta \in (0, 1)$ *with*

$$\alpha(n, k, \beta) = \min_{\sigma,\delta>0}\left\{\sqrt{2}\sigma\operatorname{erfc}^{-1}\left(\frac{\delta}{k}\right) + \exp\left(\frac{k}{2n^2\sigma^2} + \sqrt{\frac{2k}{n^2\sigma^2}\log\left(\frac{\sqrt{\pi}k}{\sqrt{2}n\sigma\delta}\right)}\right) - 1 + 6\frac{\delta}{\beta}\right\}.$$

We note that an improved bound can be obtained by (1) not constraining $\beta' = \delta$ and $c = d$ in the proof and (2) using a tighter conversion from zCDP to approximate DP based on Canonne, Kamath, and Steinke [38] in place of Bun and Steinke [37]. This improved bound is used in the approach labeled **JLNRSS+** in Section 4.2.

We now turn to the batched setting described in Section 4.2. Recall that the data arrives in $b$ batches, which we index by $\ell \in [\![b]\!]$. We let $n_\ell$ denote the size of the $\ell$-th data batch, and $k_\ell$ denote the number of queries asked by the analyst after the batch arrives. Since the $\ell$-th data batch is static while the analyst asks the $k_\ell$ queries, we can bound the worst-case distributional accuracy for the entire analysis by applying a static bound on each batch, and taking the union bound.

**Proposition C.4.** *Consider ADA on growing data in the batched setting. The clipped Gaussian mechanism can be used to answer* $k$ *statistical queries while satisfying* $(\alpha', \beta')$*-distributional accuracy for any* $\beta' \in (0, 1)$ *with* $\alpha' = \max_{\ell \in [\![b]\!]} \alpha(n_\ell, k_\ell, \beta'/b)$, *where* $\alpha(\cdot, \cdot, \cdot)$ *is as defined in Theorem C.3.*

*Proof.* Let $\boldsymbol{X} \sim \mathcal{P}^n$ and $\Pi \sim \mathsf{I}(\boldsymbol{X}; \mathsf{A}, \mathsf{M})$. It is convenient to refer to the queries and responses in the transcript $\Pi$ using two indices: the batch $\ell \in [\![b]\!]$ when the query was asked and the number $j \in [\![k_\ell]\!]$ of the query within batch $\ell$. Adopting these indices, we say that the clipped Gaussian mechanism is $(\alpha', \beta')$-distributionally accurate if with probability $1 - \beta'$ over the randomness in $\boldsymbol{X}$ and $\Pi$, we have that $\max_{\ell \in [\![b]\!]} \max_{j \in [\![k_\ell]\!]} |R_{\ell,j} - Q_{\ell,j}(\mathcal{P}^{n_\ell})| \geq \alpha'$ for any data distribution $\mathcal{P}$ and analyst $\mathsf{A}$.

Now we obtain an upper bound on the probability of interest by replacing the error $\alpha'$ with a lower bound for each batch $\ell$, and finally applying the union bound:

$$
\mathop{\mathbb{P}}_{\boldsymbol{X}\sim\mathcal{P}^n,\Pi\sim\mathsf{I}(\boldsymbol{X})}\left(\max_{\ell\in[\![b]\!]}\max_{j\in[\![k_\ell]\!]}|R_{\ell,j}-Q_{\ell,j}(\mathcal{P}^{n_\ell})|\geq\max_{\ell'\in[\![b]\!]}\alpha_{n_{\ell'},k_{\ell'}}\right)
$$

$$
=\mathop{\mathbb{P}}_{\boldsymbol{X}\sim\mathcal{P}^n,\Pi\sim\mathsf{I}(\boldsymbol{X})}\left(\max_{\ell\in[\![b]\!]}\left\{\max_{j\in[\![k_\ell]\!]}|R_{\ell,j}-Q_{\ell,j}(\mathcal{P}^{n_\ell})|-\max_{\ell'\in[\![b]\!]}\alpha_{n_{\ell'},k_{\ell'}}\right\}\geq0\right)
$$

$$
\leq\mathop{\mathbb{P}}_{\boldsymbol{X}\sim\mathcal{P}^n,\Pi\sim\mathsf{I}(\boldsymbol{X})}\left(\max_{\ell\in[\![b]\!]}\left\{\max_{j\in[\![k_\ell]\!]}|R_{\ell,j}-Q_{\ell,j}(\mathcal{P}^{n_\ell})|-\alpha_{n_\ell,k_\ell}\right\}\geq0\right)
$$

$$
=\mathop{\mathbb{P}}_{\boldsymbol{X}\sim\mathcal{P}^n,\Pi\sim\mathsf{I}(\boldsymbol{X})}\left(\bigcup_{\ell=1}^{b}\left\{\max_{j\in[\![k_\ell]\!]}|R_{\ell,j}-Q_{\ell,j}(\mathcal{P}^{n_\ell})|\geq\alpha_{n_\ell,k_\ell}\right\}\right)
$$

$$
\leq b(\beta'/b).
$$

$\square$

## C.4 Generalization Guarantees for Data Splitting

Data splitting is a simple method for mitigating the risk of overfitting when conducting an adaptive analysis. It involves randomly splitting an i.i.d. dataset into disjoint chunks, and using a fresh chunk whenever a step in the analysis depends on existing data. Data splitting has been used as a baseline in prior work for static data [10, 14, 15] and can be adapted for growing data by splitting the data into chunks as data points arrive. There is a limitation in the growing setting: it may be necessary to delay a step in the analysis if sufficient data has not yet arrived to create a fresh chunk of the desired size. This is not a limitation of our approach (Algorithm 1) which can respond to queries without delay at any time. For completeness, we provide a high probability worst-case generalization bound for data splitting below.

**Proposition C.5.** *Data splitting is $(\alpha,\beta)$-distributionally accurate when used to answer $k$ adaptive statistical queries for any $\alpha,\beta\geq0$ such that $\sum_{j=1}^{k}\mathrm{e}^{-2b_j\alpha^2}=\frac{\beta}{2}$, where $n_j$ is the (predetermined) size of the split used to answer the $j$-th query. In particular, if a dataset of size $n$ is split evenly across the $k$ queries so that $n_j=n/k\in\mathbb{N}$ then $n=\frac{k}{2\alpha^2}\log\frac{2k}{\beta}$.*

*Proof.* We write $(q_j,r_j)$ to refer to the $j$-th query-estimate pair in the flattened transcript. We also define $m_j=\sum_{j'=1}^{j-1}n_{j'}$ to be the number of data points used by the mechanism prior to answering the $j$-th query. By the union bound and Hoeffding's bound we have

$$
\mathop{\mathbb{P}}_{\boldsymbol{X}\sim\mathcal{P}^n,\Pi\sim\mathsf{I}(\boldsymbol{X})}\left(\bigcup_{j=1}^{k}\{|r_j-q_j(\mathcal{P})|\geq\alpha\}\right)
$$

$$
=\mathop{\mathbb{P}}_{\boldsymbol{X}\sim\mathcal{P}^n,\Pi\sim\mathsf{I}(\boldsymbol{X})}\left(\bigcup_{j=1}^{k}\left\{\left|\frac{1}{n_j}\sum_{t=m_j}^{m_j+n_j}q_j(X_t)-\mathop{\mathbb{E}}_{X\sim\mathcal{P}}[q_j(X)]\right|\geq\alpha\right\}\right)
$$

$$
\leq\sum_{j=1}^{k}\mathop{\mathbb{P}}_{\boldsymbol{X}\sim\mathcal{P}^n,\Pi\sim\mathsf{I}(\boldsymbol{X})}\left(\left|\frac{1}{n_j}\sum_{t=m_j}^{m_j+n_j}\left\{q_j(X_t)-\mathop{\mathbb{E}}_{X\sim\mathcal{P}}[q_j(X)]\right\}\right|\geq\alpha\right)
$$

$$
\leq\sum_{j=1}^{k}2\mathrm{e}^{-2b_j\alpha^2}
$$

$\square$

## D Results for Non-uniform Privacy Parameters

In this appendix, we prove key results for differential privacy with non-uniform privacy parameters. Although many of the results are identical to the uniform case, we were unable to find proofs in

the literature. We begin by extending the definitions of approximate differential privacy (approx. DP) and zero-concentrated differential privacy (zCDP) to the non-uniform setting. Then we prove composition and post-processing for these definitions in Appendix D.1, and a conversion theorem from zCDP to approx. DP in Appendix D.2.

Informally, differential privacy (DP) is a bound on how distinguishable the outputs of a randomized algorithm will be when run on two neighboring datasets. Ordinarily, the bound on distinguishability holds uniformly over all neighboring datasets, which means the level of privacy is the same for all records in the dataset. However, in some scenarios it may be tolerable for the privacy guarantee to vary non-uniformly over records—e.g., where individuals have different privacy preferences, or where privacy is permitted to decay as records age. Non-uniform privacy definitions have been studied using pure DP as a foundation, which is known as *personalized DP* [31, 34]. Below, we extend this idea to approximate DP, by upgrading the $\delta$ parameter from a constant to a function $\boldsymbol{\delta}(\cdot)$ that varies for each record index.

**Definition D.1** (Approximate DP). Let $\epsilon \geq 0$ and $\boldsymbol{\delta} \colon [\![n]\!] \to [0,1]$. A randomized mechanism $\mathsf{M} \colon \mathcal{X}^n \to \mathcal{Y}$ satisfies $(\epsilon, \boldsymbol{\delta})$-differential privacy or $(\epsilon, \boldsymbol{\delta})$-DP for short, if for all indices $i \in [\![n]\!]$, all pairs of neighboring datasets $(\boldsymbol{x}, \boldsymbol{x}') \in \mathcal{N}_i$ differing on the $i$-th entry, and all measurable events $E \subseteq \mathcal{Y}$:
$$\mathbb{P}(\mathsf{M}(\boldsymbol{x}) \in E) \leq \mathsf{e}^\epsilon \, \mathbb{P}(\mathsf{M}(\boldsymbol{x}') \in E) + \boldsymbol{\delta}(i).$$

Note that this definition depends on $\mathcal{N}_i$, the set of neighboring datasets that differ on the $i$-th record. One could consider *unbounded* neighboring datasets, in which case $\mathcal{N}_i$ would consist of pairs of datasets $(\boldsymbol{x}, \boldsymbol{x}')$ such that $\boldsymbol{x}'$ can be obtained from $\boldsymbol{x}$ by adding or removing the $i$-th record. Alternatively, one could consider *bounded* neighboring datasets, where the pairs of datasets $(\boldsymbol{x}, \boldsymbol{x}')$ in $\mathcal{N}_i$ are such that $\boldsymbol{x}'$ can be obtained from $\boldsymbol{x}$ by changing the $i$-th record.

Next, we define a non-uniform variant of zero-concentrated differential privacy (zCDP). However, we must first define the privacy loss distribution, since it is used to measure indistinguishability for zCDP.

**Definition D.2** (Privacy loss distribution). Let $P$ and $Q$ be probability distributions on $\mathcal{Y}$.[7] Define $f_{P\|Q} \colon \mathcal{Y} \to \mathbb{R}$ such that $f_{P\|Q}(y) = \log(P(y)/Q(y))$. The privacy loss random variable is given by $Z = f_{P\|Q}(Y)$ for $Y \leftarrow P$. The distribution of $Z$ is denoted by $\mathrm{PrivLoss}(P\|Q)$.

The standard definition of zCDP bounds the moment generating function of the privacy loss random variable $Z = \mathrm{PrivLoss}(\mathsf{M}(\boldsymbol{x})\|\mathsf{M}(\boldsymbol{x}'))$ uniformly over pairs of neighboring datasets, in terms of a scalar $\rho$ [37]. We upgrade the scalar $\rho$ to a function $\boldsymbol{\rho}(\cdot)$ that varies for each record index.

**Definition D.3** (Zero-concentrated DP). Let $\boldsymbol{\rho} \colon [\![n]\!] \to [0, \infty)$. A randomized mechanism $\mathsf{M} \colon \mathcal{X}^n \to \mathcal{Y}$ satisfies $\boldsymbol{\rho}$-zero-concentrated differential privacy or $\boldsymbol{\rho}$-zCDP for short, if for all indices $i \in [\![n]\!]$ and all pairs of neighboring datasets $(\boldsymbol{x}, \boldsymbol{x}') \in \mathcal{N}_i$ that differ on the $i$-th entry, the privacy loss distribution $\mathrm{PrivLoss}(\mathsf{M}(\boldsymbol{x})\|\mathsf{M}(\boldsymbol{x}'))$ is well-defined and
$$\forall \tau \geq 0, \; \mathbb{E}_{Z \leftarrow \mathrm{PrivLoss}(\mathsf{M}(\boldsymbol{x})\|\mathsf{M}(\boldsymbol{x}'))} (\exp(\tau Z)) \leq \exp(\tau(\tau + 1)\boldsymbol{\rho}(i)).$$

## D.1 Composition and post-processing

We need a composition theorem for $\boldsymbol{\rho}$-zCDP to analyze the privacy of successive applications of the clipped Gaussian mechanism in Lemma 4.2. We show that $\boldsymbol{\rho}$-zCDP composes in the obvious way: by adding the $\boldsymbol{\rho}$ privacy parameters pointwise.

**Theorem D.4** (Composition for $\boldsymbol{\rho}$-zCDP). *Let randomized mechanism $\mathsf{M}_1 : \mathcal{X}^\star \to \mathcal{Y}_1$ satisfy $\boldsymbol{\rho}_1$-zCDP. Let $\mathsf{M}_2 : \mathcal{X}^\star \times \mathcal{Y}_1 \to \mathcal{Y}_2$ be such that, for all $y \in \mathcal{Y}_1$, the restriction $\mathsf{M}_2(\cdot, y) : \mathcal{X}^\star \to \mathcal{Y}_2$ satisfies $\boldsymbol{\rho}_2$-zCDP. Define $\mathsf{M} : \mathcal{X}^\star \to \mathcal{Y}_1 \times \mathcal{Y}_2$ such that $Y_1 \leftarrow \mathsf{M}_1(\boldsymbol{x})$, $Y_2|Y_1 \leftarrow \mathsf{M}_2(\boldsymbol{x}, Y_1)$ and $\mathsf{M}(\boldsymbol{x}) = (Y_1, Y_2)$. Then $\mathsf{M}$ satisfies $\boldsymbol{\rho}$-zCDP with $\boldsymbol{\rho}(i) = \boldsymbol{\rho}_1(i) + \boldsymbol{\rho}_2(i)$ for all $i \in [\![n]\!]$.*

*Proof.* We adapt the proof of Steinke [55]. Fix $i \in [\![n]\!]$ and neighboring datasets $(\boldsymbol{x}, \boldsymbol{x}') \in \mathcal{N}_i$ that differ on the $i$-th entry. Fix $\tau \geq 0$. Let $Z \leftarrow \mathrm{PrivLoss}(\mathsf{M}(\boldsymbol{x})\|\mathsf{M}(\boldsymbol{x}'))$ where we conceal the dependence on $i$. We must prove $\mathbb{E}(\exp(\tau Z)) = \exp(\tau(\tau + 1)(\rho_1(i) + \rho_2(i)))$.

---

[7] For simplicity we assume $\mathcal{Y}$ is discrete, so that we don't have to worry about measure theory.

The privacy loss distribution for M can be decomposed as follows[8]:

$$f_{\mathsf{M}(\boldsymbol{x})\|\mathsf{M}(\boldsymbol{x}')}(y_1, y_2) = \log \frac{\mathbb{P}(\mathsf{M}(\boldsymbol{x}) = (y_1, y_2))}{\mathbb{P}(\mathsf{M}(\boldsymbol{x}') = (y_1, y_2))}$$

$$= \log \frac{\mathbb{P}(\mathsf{M}_1(\boldsymbol{x}) = y_1)\, \mathbb{P}(\mathsf{M}_2(\boldsymbol{x}, y_1) = y_2)}{\mathbb{P}(\mathsf{M}_1(\boldsymbol{x}') = y_1)\, \mathbb{P}(\mathsf{M}_2(\boldsymbol{x}', y_1) = y_2)}$$

$$= \log \frac{\mathbb{P}(\mathsf{M}_1(\boldsymbol{x}) = y_1)}{\mathbb{P}(\mathsf{M}_1(\boldsymbol{x}') = y_1)} + \log \frac{\mathbb{P}(\mathsf{M}_2(\boldsymbol{x}, y_1) = y_2)}{\mathbb{P}(\mathsf{M}_2(\boldsymbol{x}', y_1) = y_2)}$$

$$= f_{\mathsf{M}_1(\boldsymbol{x})\|\mathsf{M}_1(\boldsymbol{x}')}(y_1) + f_{\mathsf{M}_2(\boldsymbol{x}, y_1)\|\mathsf{M}_2(\boldsymbol{x}', y_1)}(y_2).$$

Hence

$$\mathop{\mathbb{E}}_{Z \leftarrow \mathrm{PrivLoss}(\mathsf{M}(\boldsymbol{x})\|\mathsf{M}(\boldsymbol{x}'))} (\exp(\tau Z))$$

$$= \mathop{\mathbb{E}}_{Y_1 \leftarrow \mathsf{M}_1(\boldsymbol{x}), Y_2 \leftarrow \mathsf{M}_2(\boldsymbol{x}, Y_1)} \left( \exp(\tau f_{\mathsf{M}(\boldsymbol{x})\|\mathsf{M}(\boldsymbol{x}')}(Y_1, Y_2)) \right)$$

$$= \mathop{\mathbb{E}}_{Y_1 \leftarrow \mathsf{M}_1(\boldsymbol{x})} \left( \exp(\tau f_{\mathsf{M}_1(\boldsymbol{x})\|\mathsf{M}_1(\boldsymbol{x}')}(Y_1)) \mathop{\mathbb{E}}_{Y_2 \leftarrow \mathsf{M}_2(\boldsymbol{x}, Y_1)} (\exp(\tau f_{\mathsf{M}_2(\boldsymbol{x})\|\mathsf{M}_2(\boldsymbol{x}')}(Y_2))) \right)$$

$$\leq \mathop{\mathbb{E}}_{Y_1 \leftarrow \mathsf{M}_1(\boldsymbol{x})} \left( \exp(\tau f_{\mathsf{M}_1(\boldsymbol{x})\|\mathsf{M}_1(\boldsymbol{x}')}(Y_1)) \sup_{y_1 \in \mathcal{Y}_1} \mathop{\mathbb{E}}_{Y_2 \leftarrow \mathsf{M}_2(\boldsymbol{x}, y_1)} (\exp(\tau f_{\mathsf{M}_2(\boldsymbol{x})\|\mathsf{M}_2(\boldsymbol{x}')}(Y_2))) \right)$$

$$= \mathop{\mathbb{E}}_{Z_1 \leftarrow \mathrm{PrivLoss}(\mathsf{M}_1(\boldsymbol{x})\|\mathsf{M}_1(\boldsymbol{x}'))} (\exp(\tau Z_1)) \cdot \sup_{y_1 \in \mathcal{Y}_1} \mathop{\mathbb{E}}_{Z_2 \leftarrow \mathrm{PrivLoss}(\mathsf{M}_2(\boldsymbol{x}, y_1)\|\mathsf{M}_2(\boldsymbol{x}', y_1))} (\exp(\tau Z_2))$$

$$\leq \exp(\tau(\tau + 1)\rho_1(i)) \cdot \exp(\tau(\tau + 1)\rho_2(i))$$

$$= \exp(\tau(\tau + 1)(\rho_1(i) + \rho_2(i)))$$

as required. $\qquad\square$

We require post-processing of $\boldsymbol{\rho}$-zCDP to perform privacy accounting of the entire interaction between the analyst and mechanism in 4.2. In essence, we model the adversary as a post-processing operation applied to previous responses from the clipped Gaussian mechanism, which selects the next query. We demonstrate below that post-processing holds in the non-uniform setting.

**Theorem D.5** (Post-processing for $\boldsymbol{\rho}$-zCDP). *Let* $\mathsf{M}_0 \colon \mathcal{X}^n \to \mathcal{Y}$ *be a randomized mechanism that satisfies* $\boldsymbol{\rho}$-zCDP *and let* $f \colon \mathcal{X}^n \to \mathcal{Z}$ *be an arbitrary randomized mapping. Define the post-processed mechanism* $\mathsf{M} \colon \mathcal{X}^n \to \mathcal{Z}$ *such that* $\mathsf{M}(\boldsymbol{x}) = f(\mathsf{M}_0(\boldsymbol{x}))$. *Then* $\mathsf{M}$ *also satisfies* $\boldsymbol{\rho}$-zCDP.

*Proof.* Fix $i \in [\![n]\!]$ and neighboring datasets $(\boldsymbol{x}, \boldsymbol{x}') \in \mathcal{N}_i$ that differ on the $i$-th entry. Fix $\tau \geq 0$. Applying Lemma 20 of Steinke [55], we have

$$\mathop{\mathbb{E}}_{Z \leftarrow \mathrm{PrivLoss}(\mathsf{M}(\boldsymbol{x})\|\mathsf{M}(\boldsymbol{x}'))} (\exp(\tau Z)) \leq \mathop{\mathbb{E}}_{Z_0 \leftarrow \mathrm{PrivLoss}(\mathsf{M}_0(\boldsymbol{x})\|\mathsf{M}_0(\boldsymbol{x}'))} (\exp(\tau Z_0)).$$

Now the right-hand side is $\leq \exp(\tau(\tau + 1)\boldsymbol{\rho}(i))$ since $\mathsf{M}_0$ is $\boldsymbol{\rho}$-zCDP, as required. $\qquad\square$

We also show that post-processing holds for non-uniform approximate DP. This result is used to prove generalization guarantees for minimization queries in Theorem A.3, which can be regarded as post-processed low sensitivity queries.

**Theorem D.6** (Post-processing for $(\epsilon, \boldsymbol{\delta})$-DP). *Let* $\mathsf{M}_0 \colon \mathcal{X}^n \to \mathcal{Y}$ *be a randomized mechanism that satisfies* $(\epsilon, \boldsymbol{\delta})$-DP *and let* $f \colon \mathcal{X}^n \to \mathcal{Z}$ *be an arbitrary randomized mapping. Define the post-processed mechanism* $\mathsf{M} \colon \mathcal{X}^n \to \mathcal{Z}$ *such that* $\mathsf{M}(\boldsymbol{x}) = f(\mathsf{M}_0(\boldsymbol{x}))$. *Then* $\mathsf{M}$ *also satisfies* $(\epsilon, \boldsymbol{\delta})$-DP.

*Proof.* We adapt the proof of Dwork and Roth [56]. First consider a deterministic mapping $f$. Fix $i \in [\![n]\!]$ and neighboring datasets $(\boldsymbol{x}, \boldsymbol{x}') \in \mathcal{N}_i$ that differ on the $i$-th entry. Fix any event $E \subseteq \mathcal{O}$ and let $G = \{o \in \mathcal{O} : f(o) \in E\}$. We then have

$$\mathbb{P}(\mathsf{M}(\boldsymbol{x}) \in E) = \mathbb{P}(\mathsf{M}_0(\boldsymbol{x}) \in G)$$

$$\leq \mathsf{e}^\epsilon \, \mathbb{P}(\mathsf{M}_0(\boldsymbol{x}') \in G) + \boldsymbol{\delta}(i)$$

$$= \mathsf{e}^\epsilon \, \mathbb{P}(\mathsf{M}(\boldsymbol{x}') \in E) + \boldsymbol{\delta}(i),$$

---

[8]We again assume that $\mathcal{Y}$ is discrete for simplicity, however the result holds more generally.

which completes the proof for a deterministic mapping. To extend the result to a random mapping, we can write $f$ as a mixture of deterministic mappings. The result then follows, since a mixture of $(\epsilon, \boldsymbol{\delta})$-DP mechanisms is also $(\epsilon, \boldsymbol{\delta})$-DP. □

## D.2  Converting zero-concentrated DP to approximate DP

It is convenient to analyze the privacy of the interaction $\mathsf{I}(\boldsymbol{X}; \mathsf{A}, \mathsf{M})$ for the clipped Gaussian mechanism using $\boldsymbol{\rho}$-zCDP, since it provides tighter accounting than $(\epsilon, \boldsymbol{\delta})$-DP. However, we ultimately need to convert to $(\epsilon, \boldsymbol{\delta})$-DP in order to obtain a generalization guarantee using Theorem 3.4. Canonne, Kamath, and Steinke [38] provide a conversion result from $\boldsymbol{\rho}$-zCDP to $(\epsilon, \delta)$-DP when the privacy parameters are uniform. Here we generalize their result to non-uniform privacy parameters.

We begin by generalizing Lemma 9 of Canonne, Kamath, and Steinke [38]. This is a technical result that lower bounds $\boldsymbol{\delta}(i)$ in terms of an expectation involving the privacy loss random variable.

**Lemma D.7.** *Let* $\epsilon \geq 0$ *and* $\boldsymbol{\delta} \colon [\![n]\!] \to [0, \infty)$. *A randomized mechanism* $\mathsf{M} \colon \mathcal{X}^\star \to \mathcal{Y}$ *satisfies* $(\epsilon, \boldsymbol{\delta})$-*DP if and only if*

$$\boldsymbol{\delta}(i) \geq \mathop{\mathbb{E}}_{Z_i \leftarrow \mathrm{PrivLoss}(\mathsf{M}(\boldsymbol{x}) \| \mathsf{M}(\boldsymbol{x}'))} \left( \max \left\{ 0, 1 - \mathsf{e}^{\epsilon - Z_i} \right\} \right)$$

*for all indices* $i \in [\![n]\!]$ *and neighboring datasets* $(\boldsymbol{x}, \boldsymbol{x}') \in \mathcal{N}_i$.

*Proof.* Fix $i \in [\![n]\!]$ and neighboring datasets $(\boldsymbol{x}, \boldsymbol{x}') \in \mathcal{N}_i$. Let $Z_i \leftarrow \mathrm{PrivLoss}(\mathsf{M}(\boldsymbol{x}) \| \mathsf{M}(\boldsymbol{x}'))$ and $Z_i' \leftarrow \mathrm{PrivLoss}(\mathsf{M}(\boldsymbol{x}') \| \mathsf{M}(\boldsymbol{x}))$. Our goal is to prove that

$$\sup_{E \subseteq \mathcal{Y}} \mathbb{P}(\mathsf{M}(\boldsymbol{x}) \in E) - \mathsf{e}^\epsilon \, \mathbb{P}(\mathsf{M}(\boldsymbol{x}') \in E) = \mathbb{E}(\max\{0, 1 - \mathsf{e}^{\epsilon - Z_i}\}).$$

For any $E \subseteq \mathcal{Y}$, we have

$$\mathbb{P}(\mathsf{M}(\boldsymbol{x}') \in E) = \mathbb{E}\left( \mathbf{1}_{[\mathsf{M}(\boldsymbol{x}') \in E]} \right) = \mathbb{E}\left( \mathbf{1}_{[\mathsf{M}(\boldsymbol{x}) \in E]} \mathsf{e}^{-Z_i} \right).$$

Thus for all $E \subseteq \mathcal{Y}$, we have

$$\mathbb{P}(\mathsf{M}(\boldsymbol{x}) \in E) - \mathsf{e}^\epsilon \, \mathbb{P}(\mathsf{M}(\boldsymbol{x}') \in E) = \mathbb{E}\left( \mathbf{1}_{[\mathsf{M}(\boldsymbol{x}) \in E]} \right) - \mathsf{e}^\epsilon \, \mathbb{E}\left( \mathbf{1}_{[\mathsf{M}(\boldsymbol{x}) \in E]} \mathsf{e}^{-Z_i} \right)$$
$$= \mathbb{E}\left( \mathbf{1}_{[\mathsf{M}(\boldsymbol{x}) \in E]} (1 - \mathsf{e}^{\epsilon - Z_i}) \right).$$

The worst event is $E = \{y \in \mathcal{Y} : 1 - \mathsf{e}^{\epsilon - Z_i} > 0\}$. Thus

$$\sup_{E \subseteq \mathcal{Y}} \mathbb{P}(\mathsf{M}(\boldsymbol{x}) \in E) - \mathsf{e}^\epsilon \, \mathbb{P}(\mathsf{M}(\boldsymbol{x}') \in E) = \mathbb{E}\left( \mathbf{1}_{[1 - \mathsf{e}^{\epsilon - Z_i} > 0]} (1 - \mathsf{e}^{\epsilon - Z_i}) \right)$$
$$= \mathbb{E}(\max\{0, 1 - \mathsf{e}^{\epsilon - Z_i}\})$$

as required. □

We can then use this lemma to generalize Proposition 12 of Canonne, Kamath, and Steinke [38], which converts Rényi DP to approximate DP. This is a step towards our final goal of converting zCDP to approximate DP, since zCDP is equivalent to enforcing Rényi DP over a range of privacy parameters.

**Proposition D.8.** *Let* $\mathsf{M} \colon \mathcal{X}^\star \to \mathcal{Y}$ *be a randomized mechanism. Let* $\alpha \in (1, \infty)$ *and* $\epsilon \geq 0$. *Suppose* $\mathrm{D}_\alpha(\mathsf{M}(\boldsymbol{x}) \| \mathsf{M}(\boldsymbol{x}')) \leq \boldsymbol{\rho}(i)$ *for all indices* $i$ *and neighboring datasets* $(\boldsymbol{x}, \boldsymbol{x}') \in \mathcal{N}_i$, *where* $\mathrm{D}_\alpha(P \| Q)$ *is the Rényi divergence of order* $\alpha$ *of distribution* $P$ *from distribution* $Q$. *Then* $\mathsf{M}$ *is* $(\epsilon, \boldsymbol{\delta})$-*differentially private for*

$$\boldsymbol{\delta}(i) = \frac{\mathsf{e}^{(\alpha-1)(\boldsymbol{\rho}(i) - \epsilon)}}{\alpha - 1} \left( 1 - \frac{1}{\alpha} \right)^\alpha.$$

*Proof.* Fix index $i \in [\![n]\!]$, neighboring datasets $(\boldsymbol{x}, \boldsymbol{x}') \in \mathcal{N}_i$ and let $Z_i \leftarrow \mathrm{PrivLoss}(\mathsf{M}(\boldsymbol{x}) \| \mathsf{M}(\boldsymbol{x}'))$. By assumption we have

$$\mathbb{E}\left( \mathsf{e}^{(\alpha-1)Z_i} \right) = \mathsf{e}^{(\alpha-1)\,\mathrm{D}_\alpha\left( \mathsf{M}(\boldsymbol{x}) \| \mathsf{M}(\boldsymbol{x}') \right)} \leq \mathsf{e}^{(\alpha-1)\boldsymbol{\rho}(i)}.$$

By Lemma D.7, we seek an upper bound on $\mathbb{E}(\max\{0, 1 - \mathsf{e}^{\epsilon - Z_i}\})$, which we can set to $\boldsymbol{\delta}(i)$. Following Canonne, Kamath, and Steinke, we pick $c > 0$ such that $\max\{0, 1 - \mathsf{e}^{\epsilon - z}\} \leq c \cdot \mathsf{e}^{(\alpha - 1)z}$ for all $z \in \mathbb{R}$. Then

$$\mathbb{E}\big(\max\{0, 1 - \mathsf{e}^{\epsilon - Z_i}\}\big) \leq \mathbb{E}\Big(c \cdot \mathsf{e}^{(\alpha - 1)Z_i}\Big) \leq c \cdot \mathsf{e}^{(\alpha - 1)\boldsymbol{\rho}(i)}.$$

Canonne, Kamath, and Steinke show that the smallest value of $c$ that satisfies this condition is $c = \frac{\mathsf{e}^{\epsilon(1 - \alpha)}}{\alpha - 1}\left(1 - \frac{1}{\alpha}\right)^\alpha$. Thus

$$\mathbb{E}\big(\max\{0, 1 - \mathsf{e}^{\epsilon - Z_i}\}\big) \leq \frac{\mathsf{e}^{\epsilon(1 - \alpha)}}{\alpha - 1}\left(1 - \frac{1}{\alpha}\right)^\alpha \cdot \mathsf{e}^{(\alpha - 1)\boldsymbol{\rho}(i)} = \frac{\mathsf{e}^{(\alpha - 1)(\boldsymbol{\rho}(i) - \epsilon)}}{\alpha - 1}\left(1 - \frac{1}{\alpha}\right)^\alpha = \boldsymbol{\delta}(i).$$

$\square$

By exploiting the connection between Rényi DP and zCDP, we obtain a conversion result from zCDP to approximate DP. This is a generalization of Corollary 13 of Canonne, Kamath, and Steinke [38].

**Corollary D.9.** *Let* $\mathsf{M}\colon \mathcal{X}^\star \to \mathcal{Y}$ *be a randomized mechanism that satisfies* $\boldsymbol{\rho}$-*zCDP. Then* $\mathsf{M}$ *is* $(\epsilon, \boldsymbol{\delta})$-*DP for any* $\epsilon \geq 0$ *and* $\boldsymbol{\delta}: [\![n]\!] \to [0, \infty)$ *such that*

$$\boldsymbol{\delta}(i) = \frac{\mathsf{e}^{(\alpha^\star - 1)(\alpha^\star \boldsymbol{\rho}(i) - \epsilon)}}{\alpha^\star - 1}\left(1 - \frac{1}{\alpha^\star}\right)^{\alpha^\star},$$

*with* $\alpha^\star = \arg\min_{\alpha \in (1, \infty)} \frac{\mathsf{e}^{(\alpha - 1)(\alpha \sup_i \boldsymbol{\rho}(i) - \epsilon)}}{\alpha - 1}\left(1 - \frac{1}{\alpha}\right)^\alpha$.

*Proof.* Fix $i \in [\![n]\!]$ and $(\boldsymbol{x}, \boldsymbol{x}') \in \mathcal{N}_i$. Since $\mathsf{M}$ satisfies $\boldsymbol{\rho}$-zCDP, we have $\mathrm{D}_\alpha(\mathsf{M}(\boldsymbol{x}) \| \mathsf{M}(\boldsymbol{x}')) \leq \alpha\boldsymbol{\rho}(i)$ for all $\alpha \in (1, \infty)$. Proposition D.8 (with $\boldsymbol{\rho}(i) \leftarrow \alpha\boldsymbol{\rho}(i)$) provides a conversion result to $(\epsilon, \boldsymbol{\delta})$-DP for any choice of $\alpha$. We choose $\alpha$ to minimize the worst case $\boldsymbol{\delta}(i)$, given by:

$$\sup_i \boldsymbol{\delta}(i) = \frac{\mathsf{e}^{(\alpha - 1)(\alpha \sup_i \boldsymbol{\rho}(i) - \epsilon)}}{\alpha - 1}\left(1 - \frac{1}{\alpha}\right)^\alpha = \exp g(\alpha)$$

with $g(\alpha) = (\alpha - 1)(\alpha \sup_i \boldsymbol{\rho}(i) - \epsilon) + \alpha \log(1 - 1/\alpha) - \log(\alpha - 1)$. A unique minimizer $\alpha^\star$ exists since $g(\alpha)$ is a smooth convex function. $\square$

We note that the optimizer $\alpha^\star$ can be found efficiently using binary search as described by Canonne, Kamath, and Steinke [38].

# E   Additional Empirical Results

In this appendix, we provide additional empirical results to complement those presented in Section 4.2. In Figure 3 of the main paper, we plotted a lower bound on the number of adaptive statistical queries $k$ that can be answered as a function of the final dataset size $n$. There we varied the initial size of the dataset $n_0$ to ensure a fixed growth ratio $n/n_0 = 3$. In Figure 4, we produce a similar plot where we fix $n_0 = 500\,000$ and vary the growth ratio $n/n_0$ instead. Here again, we observe that our bounds (**Ours-U** and **Ours-N**) outperform the others in the non-asymptotic regime plotted, with the performance gap becoming more pronounced for larger values of $b$.

Figure 5 covers the same setting as Figure 3 in the main paper, except that it plots the confidence width $\alpha$ on the vertical axis for a fixed number of queries $k = 10\,000$. A smaller confidence width is better, and we see that the relative rankings of the bounds is the same as in Figure 3. Notably, the behavior of our bounds (**Ours-U** and **Ours-N**) is stable for all values of $b > 1$, whereas the confidence width degrades for the static-based bounds (**JLNRSS** and **JLNRSS+**) as $b$ increases.

Figure 6 examines the impact of query batching for a fixed final dataset size $n = 1\,500\,000$ and fixed initial dataset size $n_0 = 500\,000$. It shows that both of our bounds (**Ours-U** and **Ours-N**) improve as the number of batches $b$ increases, reaching a saturation point at around $b = 40$. This suggests it is better from a generalization perspective to ask queries more frequently in smaller batches when using our bounds. In contrast, the static-based bounds (**JLNRSS** and **JLNRSS+**) degrade as the number of batches $b$ increases.

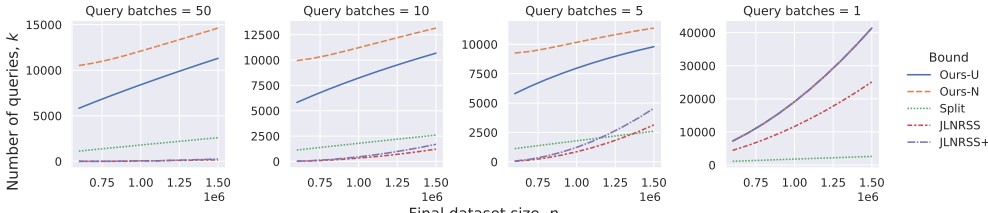

Figure 4: Comparison of the number of adaptive statistical queries that can be answered with error tolerance $\alpha = 0.1$ and uniform coverage probability $1 - \beta = 0.95$ using a growing dataset with fixed initial size $n_0 = 500\,000$ in a batched query setting. The number of queries (vertical axis) is plotted as a function of the final dataset size $n$ (horizontal axis), bound (curve style) and the number of query batches $b$ (horizontal panel). The right-most panel with $b = 1$ corresponds to the static data setting.

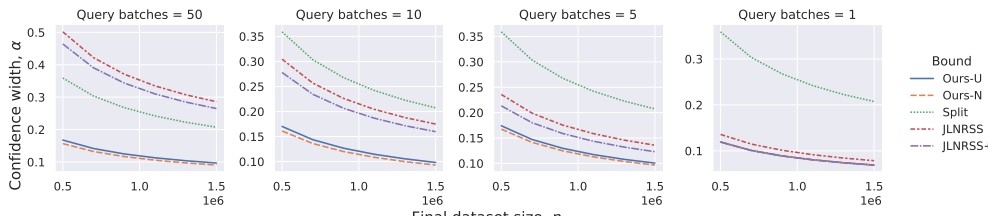

Figure 5: Comparison of the error tolerance $\alpha$ (vertical axis) for $k = 10\,000$ adaptive statistical queries with uniform coverage probability $1 - \beta = 0.95$ using different ADA bounds (curve style). The queries are answered using a growing dataset with final size $n$ (horizontal axis) and growth ratio $n/n_0 = 3$, and are grouped into $b$ batches (horizontal panel). The right-most panel with $b = 1$ corresponds to the static data setting.

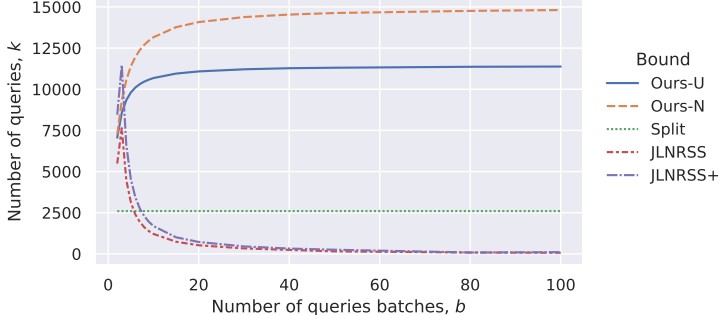

Figure 6: Comparison of the number of adaptive statistical queries that can be answered with error tolerance $\alpha = 0.1$ and uniform coverage probability $1 - \beta = 0.95$ using a growing dataset with initial size $n_0 = 500\,000$ and final size $n = 1\,500\,000$ in a batched query setting. The number of queries (vertical axis) is plotted as a function of the number of query batches $b$ (horizontal axis) and bound (curve style).

