# OpenReview forum: "Adaptive Data Analysis for Growing Data"
_NeurIPS.cc/2025/Conference — NeurIPS 2025 poster_

### Official Review · Reviewer_BVSk · 2025-06-24

**Clarity:** 4
**Significance:** 3
**Originality:** 2
**Rating:** 4
**Confidence:** 4

**Summary:**

As suggested by its title, this paper considers adaptive data analysis (ADA) - an interaction between a data curator and an analyst(s) issuing adaptive queries based on previous responses, where data is accumulated throughout the interaction. Previous work proved that ADA may lead to statistically invalid results by letting the analyst "overfit" the queries to the dataset, even without direct access to that data, just by observing responses to past queries. This risk can be mitigated by ensuring some stability properties of the response producing mechanism, such as the ones guaranteed by differential privacy (DP), which guarantee statistical validity for $O(n^2)$ queries where $n$ is the sample size.
Previous work considered only a fixed dataset setting, but in some cases it might be useful to gradually increase the dataset (e.g., as a price for the queries [1]). A natural baseline option can be fixing some $n_0$, using the first $n_0$ data elements to reply to some $O(n_{0}^{2})$ queries, then tossing those points and using the next $n_{0}$ data elements to reply to the next$O(n_{0}^{2})$ queries, and so on.
This work considers a more sophisticated technique, extending the analysis of [2] to the dynamic sample size setting, significantly increasing the number of queries that can be estimated.

[1] Woodworth Blake E. et al. "The everlasting database: Statistical validity at a fair price" (NeurIPS 2018)

[2] Christopher Jung et al. “A New Analysis of Differential Privacy’s Generalization Guarantees” (ITCS 2020)

**Questions:**

What is the purpose of the $n_0$ variable in algorithm 1?
Since the choice when to issue the queries is governed by the analyst, why can't the mechanism simply reply from $n_0=1$?

Why was $n_0$ fixed for all experiments to be $n/3$?
It seems like a fair comparison to JLNRSS would be setting $n_0 = n/b$. I assume it won't induce a huge change (equivalent to $\approx 2/3$ increase in sample size), but nevertheless it seems like the natural choice, which keeps the sample size constant throughout the rounds.

How is it possible that the empirical results in Figure 1 show the maximal number of queries that can be answered by your method for a given sample size (say $1.5 \cdot 10^6$) *decreases* with $b$ from 1 to 5 then *increase* from 5 to 10 to 50?
The existing formal guarantees are too convoluted for me to analyze asymptotically, but I would expect it to decrease with $b$ as does the baseline method of JLNRSS. This result is emphasized in Figure 6 but without explanation. Do you expect the saturation to remain even when $b \rightarrow k$? this sounds unreasonable to me.
Figure 5 presents a different surprising result. It seems like your method is barely affected by $b$. That is, the $\alpha(n)$ profile for fixed $k$ and $\beta$ barely changes as we vary $b$, which I have a hard time understanding.


**Minor comments**
* The $\Delta_t$ variables seem redundant, and can be fixed to some single $\Delta$ by resealing the queries and the $\alpha_t$ accuracy requirements accordingly, this will also simplify notations in Lemma 3.5
* The posterior distribution mentioned first in line 169 is not define in the body of the paper, and might be hard to understand for the unfamiliar reader
* It seems like the $\mathcal{N}_{t}$ notation in Definition 2.4 was not defined, and I suspect it is a mistake and should've been $\mathcal{X}^{t}$
* I think the result in Lemma 3.3 can be improved by restricting $\sum \delta(t)$ to the sum over the indexes where queries were issued (or, alternatively, we can define $\delta(t)=0$ for the rest of the rounds)
* Lemma 4.2 holds regardless of the clipping, so I recommend removing it from the statement

**Ethical Concerns:**

["NO or VERY MINOR ethics concerns only"]

**Final Justification:**

The authors provided sufficient clarification to most of my questions, though there are some empirical results which I still find less intuitive and cannot be corroborated by the theory.

**Quality:**

3

**Strengths And Weaknesses:**

**Strengths**

The main contribution of this work is the generalization of existing results in adaptive data analysis from the constant dataset size to the growing data setting. While the asymptotic (in sample size and accuracy parameters) guarantees it provides are identical to the ones provided by the natural baseline, the improvement is nevertheless significant for many finite sample sizes, and so can be proven very useful. The proof techniques, though not very different from the original work, encounter various subtle issues and iseseveral novel techniques.

**Weaknesses**

Most of the mathematical analysis is a direct generalization of the work by [1]. While it requires careful care and some novel technical steps, it does not provide additional conceptual insights.

The paper is lacking comparisons to direct application of the results for DP [2, 3], which might be relevant (though, some are computationally inefficient, so I am not certain there is a clear comparison)

I find the empirical data results somewhat concerning (see questions section). I am open to change my score if this question will be settled.

[1] Christopher Jung et al. “A New Analysis of Differential Privacy’s Generalization Guarantees” (ITCS 2020)

[2] Rachel Cummings et al. “Differential Privacy for Growing Databases” (NeurIPS 2018)

[3] Yuan Qiu and Ke Yi. "Differential Privacy on Dynamic Data" (ArXiv 2022)

---

> ### Author Rebuttal · Authors · 2025-07-31
>
> Thank you for your detailed review and for recognizing the significance of our work in generalizing existing results in adaptive data analysis to the growing data setting. Now that we have addressed your concerns, in particular your core question about empirical data results, we hope that you will update your score as offered. If you have any follow-up questions, we'd be happy to address them.
>
> > The paper is lacking comparisons to direct application of the results for DP [2, 3], which might be relevant (though, some are computationally inefficient, so I am not certain there is a clear comparison)
>
> Thank you for this suggestion. The papers you mention study DP mechanisms for growing data, providing asymptotic bounds for snapshot accuracy (a.k.a. sample accuracy). However, they don't address generalization, which is the focus of our work. While it's possible to plug in their results for snapshot accuracy and privacy into our framework, we didn't pursue this because the papers don't provide the necessary constants for a meaningful comparison.
>
> > What is the purpose of the variable $n_0$ in algorithm 1? Since the choice when to issue the queries is governed by the analyst, why can't the mechanism simply reply from $n_0 = 1$?
>
> The variable $n_0$ represents the initial sample size before the analysis begins. While our bounds can theoretically be instantiated with $n_0 = 1$, this would lead to vacuous results due to low posterior stability/privacy. This is expected behavior – it is difficult to achieve reasonable privacy on a dataset containing one record without adding a large amount of noise.
>
> > Why was $n_0$ fixed for all experiments to be $n/3$? It seems like a fair comparison to JLNRSS would be setting $n_0 = n / b$.
>
> We'd like to clarify a few points regarding $n_0$:
>
> - We don't fix $n_0 = n/3$ in all experiments. In Figs 4 and 6, we vary $n$ independently of $n_0$, allowing us to explore a range of scenarios.
>
> - The choice of $n_0 = n/3$ reflects scenarios where a substantial initial dataset is available before adaptive querying begins. This choice balances immediate utility (answering queries early) with overall performance (maintaining low error across all queries).
>
> > How is it possible that the empirical results in Figure 1 show the maximal number of queries that can be answered by your method for a given sample size (say $1.5 \cdot 10^6$) decreases with $b$  from 1 to 5 then increase from 5 to 10 to 50?
>
> We believe you're referring to Fig 3 rather than Fig 1. The behavior you've observed is due to the special case definition for $b = 1$. When $b = 1$ and $n > n_0$, we stipulate that all queries are asked in a single batch at round $n$, which is equivalent to the static setting from prior work. (Note: if we didn’t define this special behavior for $b = 1$, all queries would be asked at round $n_0$, ignoring the remaining $n - n_0$ data points, which isn’t an interesting case to consider).
>
> This explains the discontinuous behavior between $b = 1$ and $b > 1$. We can see how this is not clear from the figure’s presentation, and are considering replacing the “$b = 1$” label with “Static” instead, but we welcome your advice on how best to present this distinction.
>
> > The existing formal guarantees are too convoluted for me to analyze asymptotically, but I would expect it to decrease with $b$ as does the baseline method of JLNRSS. This result is emphasized in Figure 6 but without explanation. Do you expect the saturation to remain even when $b \to k$? this sounds unreasonable to me. Figure 5 presents a different surprising result. It seems like your method is barely affected by $b$. That is, the $\alpha(n)$ profile for fixed $k$ and $\beta$ barely changes as we vary $b$, which I have a hard time understanding.
>
> These are interesting questions. We don’t have a strong intuition for why the asymptotic behavior would disagree with our empirical results – namely that the number of queries $k$ increases with $b$ for fixed $\alpha’$, and the confidence width $\alpha’$ decreases with $b$ for fixed $k$.
>
> As you rightly point out, the bound is difficult to analyze asymptotically (in the limit $b \to n - n_0$), as it is defined as the solution to an optimization problem over the free parameters, which may depend implicitly on $b$. As requested here is a table for Fig 5.
>
> | Bound | Batches (b) | Final size (n) | Confidence width (ɑ) |
> |--------|--:|--:|------:|
> | Ours-U | 1 | 500000 | 0.119229 |
> | | | 700000 | 0.100801 |
> | | | 900000 | 0.0889361 |
> | | | 1100000 | 0.0804802 |
> | | | 1300000 | 0.0740615 |
> | | | 1500000 | 0.0689744 |
> | | 5 | 500000 | 0.174153 |
> | | | 700000 | 0.147117 |
> | | | 900000 | 0.129732 |
> | | | 1100000 | 0.117354 |
> | | | 1300000 | 0.107964 |
> | | | 1500000 | 0.100526 |
> | | 10 | 500000 | 0.170241 |
> | | | 700000 | 0.14382 |
> | | | 900000 | 0.126829 |
> | | | 1100000 | 0.114731 |
> | | | 1300000 | 0.105553 |
> | | | 1500000 | 0.0982822 |
> | | 50 | 500000 | 0.167663 |
> | | | 700000 | 0.141648 |
> | | | 900000 | 0.124917 |
> | | | 1100000 | 0.113003 |
> | | | 1300000 | 0.103964 |
> | | | 1500000 | 0.096804 |
> | Ours-N | 1 | 500000 | 0.119229 |
> | | | 700000 | 0.100801 |
> | | | 900000 | 0.0889361 |
> | | | 1100000 | 0.0804802 |
> | | | 1300000 | 0.0740615 |
> | | | 1500000 | 0.0689744 |
> | | 5 | 500000 | 0.167389 |
> | | | 700000 | 0.141418 |
> | | | 900000 | 0.124715 |
> | | | 1100000 | 0.112821 |
> | | | 1300000 | 0.103797 |
> | | | 1500000 | 0.0966487 |
> | | 10 | 500000 | 0.161115 |
> | | | 700000 | 0.136131 |
> | | | 900000 | 0.120059 |
> | | | 1100000 | 0.108614 |
> | | | 1300000 | 0.0999301 |
> | | | 1500000 | 0.0930505 |
> | | 50 | 500000 | 0.156637 |
> | | | 700000 | 0.132356 |
> | | | 900000 | 0.116736 |
> | | | 1100000 | 0.105611 |
> | | | 1300000 | 0.0971692 |
> | | | 1500000 | 0.0904813 |
>
> We observe that the error does decrease marginally with increasing $b$ which is consistent with Fig 6.
>
> > The $\Delta_t$ variables seem redundant, and can be fixed to some single $\Delta$ by resealing the queries
>
> Perhaps we’re missing something, but we can’t see how this would work. If we were to abandon the round-dependent sensitivity $\Delta_t$, we would sacrifice some tightness in the bounds. In Lemma 3.5, the posterior stability depends on $\min_t t \Delta_t$, $\max_t t \Delta_t$, and $\max_t \Delta_t$. These three factors cannot be described using a single sensitivity without sacrificing tightness.
>
> > Posterior distribution from line 169 is not defined in the body of the paper
>
> This is defined on line 172 as $\mathcal{Q}_{\Pi}^{t} = \mathcal{P}^t | \Pi$. Please let us know if this is unclear.
>
> > It seems like the $\mathcal{N}_t$ notation in Definition 2.4 was not defined, and I suspect it is a mistake and should've been $\mathcal{X}^t$.
>
> $\mathcal{N}_t$ is defined in words on lines 187-188. It denotes the set of neighboring datasets that differ on the $t$-th record. It is not equal to $\mathcal{X}^t$.
>
> > I think the result in Lemma 3.3 can be improved by restricting $\sum \delta(t)$ to the sum over the indexes where queries were issued
>
> We don’t see how this would work. In Lemma 3.3, we don’t assume knowledge of the rounds in which queries are issued. Rather, we consider a worst-case analyst. The proof is centered around the round, query and response that result in the largest error (the starred quantities in the proof). After invoking DP on a carefully constructed event (eq. 15), the sum over $\delta(t)$ appears without any connection to the transcript $\pi$ (which specifies the rounds in which queries were issued). We therefore don’t see how the sum could be restricted to only those rounds for which queries were issued.
>
> > Lemma 4.2 holds regardless of the clipping, so I recommend removing it from the statement
>
> You're correct that Lemma 4.2 holds for both the clipped and ordinary Gaussian mechanisms. We initially presented it for the clipped Gaussian mechanism as our work assumes [0, 1]-bounded queries. However, for generality, we'll revise the statement in Lemma 4.2 to read: "where $M$ is the ordinary or clipped Gaussian mechanism".

---

> ### Comment · Reviewer_BVSk · 2025-08-02
>
> Thank you for the detailed response.
>
> I found the responses to most questions satisfactory.
>
> I am still somewhat concerned with the confusion resulting from fixing $n_0=n/3$ in the experiment of Figure 3 (indeed, I meant 3, not 1). Besides my concern that it leads to a somewhat unfair comparison to JLNRSS, it also leads to an unnatural separation between the $b=1$ and $b>1$ settings. My recommendation would be to slightly change the experiment by issuing the queries at $b$ points each one after $n/b$ elements. This way the $b=1$ naturally becomes the static case.
>
> Minor questions and comments:
> 1. My  $n_0$ question was a matter of presentation. Of course, choosing to issue queries with small sample size would result in poor accuracy, but it seems like  the results can be simply expressed in terms of the choices of the analyst as with the scheduling of the rest of the queries.
> 2. My proposal to avoid the usage of $\Delta_{t}$ results from the insight that accuracy $\alpha_{t}$ is actually measured in $\Delta_{t}$ units. Consider a process where the mechanism is wrapped by a preprocess step that scaled the query such that its sensitivity is $\Delta$, and a post process step that scales back the response. In this case, all queries share the same sensitivity, and the accuracy guarantees $\alpha_t$ which apply to the rescaled queries must be rescaled as well. In any case, this is just a presentation simplification proposal, nothing that matters to the content of the work.
> 3. Regarding Lemma 3.3, I suspect it is possible to change the sum over $t$ as introduced in eq. 13 and its consequent usage to be over the indexes where queries were issued, changing the probability terms to represent the sampling of all elements sampled between rounds, but I might very well be missing something. In any case, this will probably not have a significant effect on the results.
>
> I will maintain my score, given the various phenomena in the empirical results which are not explained by the theory (though, do not contradict it, so I do not consider them a problem).

---

> > ### Author Response · Authors · 2025-08-04
> >
> > Thank you for your follow-up comments. We appreciate your continued engagement with our work.
> >
> > > Regarding Lemma 3.3, I suspect it is possible to change the sum over $t$ as introduced in eq. 13 and its consequent usage to be over the indexes where queries were issued, changing the probability terms to represent the sampling of all elements sampled between rounds, but I might very well be missing something. In any case, this will probably not have a significant effect on the results.
> >
> > We'd like to clarify that the bounds in Section 3 are worst-case with respect to the analyst, _including their query schedule_, conditioned on the interaction being $(\epsilon, \delta)$-PS or $(\epsilon, \delta)$-DP. In Section 3, we don't know the query schedule—we're considering the schedule that results in the worst-case error conditioned on the PS/DP.
> >
> > In Section 4.1, we do assume the queries follow a known schedule so that we can compute the privacy loss and snapshot accuracy, and instantiate the bounds from Section 3. However, the resulting bound could be tightened if we assumed the schedule was known from the beginning (i.e., in Section 3). We don't do this because fixing the schedule in advance would limit the analyst's freedom, and we're interested in the fully adaptive case.
> >
> > > My $n_0$ question was a matter of presentation. Of course, choosing to issue queries with small sample size would result in poor accuracy, but it seems like the results can be simply expressed in terms of the choices of the analyst as with the scheduling of the rest of the queries.
> >
> > Building on the above discussion, we hope this explains why we fix the initial dataset size $n_0$. We want to rule out worst-case analysts that ask all queries in round 1, leading to poor accuracy. We believe it is reasonable for the analyst to decide $n_0$ in advance/without looking at the data. From that point on, everything is fully adaptive.
> >
> > > I am still somewhat concerned with the confusion resulting from fixing $n_0=n/3$ in the experiment of Figure 3 (indeed, I meant 3, not 1). Besides my concern that it leads to a somewhat unfair comparison to JLNRSS, it also leads to an unnatural separation between the $b=1$ and $b>1$ settings. My recommendation would be to slightly change the experiment by issuing the queries at $b$ points each one after $n/b$ elements. This way the $b=1$ naturally becomes the static case.
> >
> > We appreciate your suggestion to divide the $b$ query batches evenly across the entire dataset. However, this scenario (with small, evenly divided batches) is not the most interesting or practical case to highlight.  For large $b$, it results in a small initial dataset size $n_0$, where all methods would struggle to provide meaningful accuracy guarantees. This explains why we set $n_0$ independently of $b$, to ensure a substantial initial dataset before adaptive querying begins.
> >
> > > My proposal to avoid the usage of $\Delta_t$ results from the insight that accuracy $\alpha_t$ is actually measured in $\Delta_t$ units. Consider a process where the mechanism is wrapped by a preprocess step that scaled the query such that its sensitivity is $\Delta$, and a post process step that scales back the response. In this case, all queries share the same sensitivity, and the accuracy guarantees which apply to the rescaled queries must be rescaled as well. In any case, this is just a presentation simplification proposal, nothing that matters to the content of the work.
> >
> > Thank you for this clarification. We understand you are proposing to state the guarantees for queries normalized to have sensitivity $\Delta$ (e.g., for simplicity one could set $\Delta=1$). While this approach has merit, it would require instantiating the guarantee based on the privacy loss for normalized queries. This might make usage of the bound more complicated, as one couldn't use privacy accounting for the mechanism that's actually used (adjustments for normalization would be necessary). We're concerned this may not simplify the presentation but rather shift complexity elsewhere. That said, we appreciate the suggestion and will give it further consideration.
> >
> > We hope these explanations address concerns and in other cases why we believe our parameterization of the setting strikes the best balance between presentation of results, and pragmatic reflection of how we believe adaptive analysis is used in practice (on an initial dataset then followed up by updates to data/analyses). We very much appreciate this stimulating conversation, and have enjoyed considering these ideas. Wherever there has been a concern raised, we believe we have fully addressed this and so we wonder if the reviewer’s score is reflective of their assessment of the work as it is currently (distinct to constructive suggestions). We’re of course delighted to further discuss alternate presentations with you during this phase, and happy to take further ideas on / discuss the previous ones further.

---

> > > ### Comment · Reviewer_BVSk · 2025-08-04
> > >
> > > I thank the authors for their clarifications.
> > >
> > > I recommend clarifying the chosen perspective of focusing on generalization guarantees against arbitrary analysts, which are not expressed in terms of their chosen schedule. At the same time, I think it would be beneficial to further pursue the option of schedule dependent generalization guarantees, and optimal scheduling under various utilities, e.g., the tradeoff (from the analyst's perspective) between getting earlier (in terms of number of points gathered) and more accurate result.
> > >
> > > Overall, this seems like a worthy work.

---

> > > > ### Author Response · Authors · 2025-08-04
> > > >
> > > > Thank you again for your valuable engagement in this discussion.
> > > >
> > > > We concur with your recommendation, and will explicitly clarify our focus on generalization guarantees against arbitrary analysts _including their querying schedule_. This clarifying change will undoubtedly help readers better understand our perspective and approach. We value your suggestion regarding schedule-dependent guarantees and optimal scheduling—these are indeed promising directions for follow up work.

---

### Official Review · Reviewer_KtCZ · 2025-07-02

**Clarity:** 3
**Significance:** 3
**Originality:** 2
**Rating:** 5
**Confidence:** 3

**Summary:**

This paper studies adaptive data analysis in the context of a growing dataset. In this line of work there is a dataset consisting of i.i.d. draws from a distribution $P$ and an analyst is interrogating it by asking an adaptively chosen sequence of queries. Our goal is to design mechanisms for responding to those queries such that, despite the data reuse, we can bound the absolute error in all query responses simultaneously with high probability. The key new aspect considered in this work is to allow the underlying dataset to grow over time as the analyst issues the queries. The specific protocol works as follows: the analyst waits for the first $n_0$ to appear and then after each successive data point arrives, the analyst can ask a sequence of queries with that snapshot of the data.

The analysis in the paper roughly follows that of Jung et al. [15]. At a high level, they argue that when a mechanism is snapshot-accurate (in the sense that every query response is close to its empirical value on the data snapshot) and the induced interaction between the analyst and mechanism is posterior-stable (in the sense that the expected query values on the posterior distribution conditioned on the transcript are not far from from the expected query distributions on the original data distribution), then it will be distributionally accurate as well (meaning that query responses are accurate for all queries). In the specific case when the Gaussian mechanism is used to answer queries, we obtain snapshot accuracy because the mechanism adds a small amount of noise to the empirical values, and the authors argue that for several query classes, differentially private mechanisms induce posterior-stable interactions. Putting this together gives generalization guarantees that hold for all the queries simultaneously.

The paper also compares empirically the number of queries that can be answered as a function of the total dataset size when the queries arrive in batches that must be answered when they were issued. For prior work that does not handle growing data, they collect a different batch of data for each batch of queries and use that static data with the prior work's method. They also consider a splitting baseline that answers each query with disjoint subset of the data. In these experiments, their proposed method is able to answer the most queries as a function of the given data size.

**Questions:**

See questions under strengths and weaknesses.

**Ethical Concerns:**

["NO or VERY MINOR ethics concerns only"]

**Final Justification:**

The authors addressed my comments adequately and after reading the other reviews I do not have major concerns, so I will keep my positive score.

**Limitations:**

Yes.

**Paper Formatting Concerns:**

None.

**Quality:**

3

**Strengths And Weaknesses:**

## Strengths
1. The paper is well written and relatively easy to follow.
1. The extension of adaptive data analysis generalization guarantees to the case where the dataset grows over time seems natural and useful in practice.
1. The proposed method performs better than natural baselines in the empirical comparison.

## Weaknesses
1. This is relatively minor, but I think the paper would be strengthened by identifying places where the analysis and arguments deviate significantly from Jung et al. [15]. After the modified definitions have been introduced, it is somewhat unclear if any of the arguments need significant new ideas, or if they more or less go through unmodified. In the introduction the authors (on line 54) write that a key insight is "in the growing data setting, [...] the posterior distribution should be marginalized over unseen future data at the time the query was submitted, while conditioning on the full history of the analysis." Is this a major change to the argument?
1. The experiments mostly focus on the setting where queries must be answered quickly and we cannot wait until the full dataset has arrived to start answering them. As an optimistic baseline, it would be interesting to include the number of queries that can be answered by prior work adaptively if we wait for the entire dataset of size n to arrive. Is it true that this number of queries should be higher than the proposed method, or do we actually gain some additional generalization due to the fact that some of the queries are answered on a subset of the data?

---

> ### Author Rebuttal · Authors · 2025-07-31
>
> We appreciate your thoughtful review and your overall positive reception of our paper. We'd like to address your comments and provide some additional context to highlight the significance of our contributions. If you find them satisfactory, we would be grateful if you would consider updating your score.
>
> > This is relatively minor, but I think the paper would be strengthened by identifying places where the analysis and arguments deviate significantly from Jung et al. [15]. After the modified definitions have been introduced, it is somewhat unclear if any of the arguments need significant new ideas, or if they more or less go through unmodified. In the introduction the authors (on line 54) write that a key insight is "in the growing data setting, [...] the posterior distribution should be marginalized over unseen future data at the time the query was submitted, while conditioning on the full history of the analysis." Is this a major change to the argument?
>
> Our approach follows a similar high-level structure to Jung et al.: upper bounding the tail probability of the distributional error using two other tail probabilities bounded by snapshot accuracy and posterior stability. However, adapting this to the growing data setting introduced challenges and novel elements.
>
> In Jung et al.'s work, the tail probabilities depend on a "reference point" for each query $q$, evaluated on the posterior distribution $\mathcal{Q}_\Pi$. For our growing data setting, we had to reconsider this reference point, choosing to marginalize over unseen future data at the time of the query. This wasn't an obvious choice and only came to us after exploration of alternatives.
>
> We then adapted the definition of posterior stability based on this new reference point. The most challenging aspect was proving the conversion lemmas from differential privacy to our new definition of posterior stability. This required careful adaptation as the support of the posterior grows with $t$. Since $\mathcal{Q}_{\Pi}^{t}$ only covers data observed up to round $t$, and the analyst controls when the mechanism ingests the next data point, the size of data used for a query depends on the transcript $\Pi$.
>
> This dependency complicated our proofs significantly. For instance, in Lemma 3.3, we could no longer interchange sums (three steps before equation 15), requiring more complex manipulations to tightly invoke differential privacy. Similar challenges arose in Lemma 3.5.
>
> We appreciate your suggestion and are happy to incorporate this discussion into the paper to clarify the distinctions between our work and previous approaches.
>
> > The experiments mostly focus on the setting where queries must be answered quickly and we cannot wait until the full dataset has arrived to start answering them. As an optimistic baseline, it would be interesting to include the number of queries that can be answered by prior work adaptively if we wait for the entire dataset of size n to arrive. Is it true that this number of queries should be higher than the proposed method, or do we actually gain some additional generalization due to the fact that some of the queries are answered on a subset of the data?
>
> In fact, our experiments do include the scenario you describe as an optimistic baseline. Specifically:
>
> - The case where the analyst waits for the entire dataset corresponds to $b = 1$ in our experiments (see line 313 in the paper).
>
> - In this case ($b = 1$), our bound and the tighter JLNRSS bound coincide, as shown in the last panel of Figure 3.
>
> - Comparing the first three panels in Figure 3 (where $b > 1$) with the last panel ($b = 1$), we can see that waiting for the full dataset allows answering roughly 3–4 times more queries, regardless of which bound is used.
>
> This comparison illustrates a key trade-off: being forced to use smaller snapshots early on does indeed harm the worst-case error over all queries. However, our method provides a way to handle scenarios where immediate query responses are necessary, while still maintaining strong generalization guarantees.

---

> > ### Comment · Reviewer_KtCZ · 2025-08-02
> > **Thanks!**
> >
> > Thank you for the clarifications on the how the arguments deviate from Jung et al. and for pointing out my misunderstanding about the experiment baselines. I will maintain my positive score.

---

### Official Review · Reviewer_ow1s · 2025-07-03

**Clarity:** 4
**Significance:** 4
**Originality:** 3
**Rating:** 5
**Confidence:** 2

**Summary:**

Differential privacy can be used to provide generalization guarantees for data workflows where queries are issued adaptively. However, all prior work assumed that the dataset is static. This paper studies adaptive data analysis when data grows over time. That is, an analyst interactively submits queries from a fixed class to a differentially private mechanism. At time $t \in [n_0, n]$, the mechanism runs only on the first t records of a "final" dataset containing n records.

The authors consider three query classes: statistical queries (standard definition), "low-sensitivity queries" (a larger class of scalar queries, where the sensitivity is a vector indexed by the dataset size) and minimization queries (solutions to certain optimization problems).

In this setting, the paper proves generalization guarantees similar to what exists in the static case. The transfer theorem says that if we have a mechanism that answers accurately based on the data available for each query (snapshot accuracy) and an interaction that satisfies DP, then the answers for each query are actually close to the the expected value of the query over the true distribution.

Finally, the bounds are instantiated by answering statistical queries with the Gaussian mechanism. The new framework outperforms a baseline that uses static ADA on disjoint data subsets for each batch of queries, and a baseline that just runs each query on a disjoint subset.

**Questions:**

How did you set the round-dependent standard deviation in the experiments, and what are the epsilon, delta (or delta(t) for Ours-N?) guarantees in Fig 3?

**Ethical Concerns:**

["NO or VERY MINOR ethics concerns only"]

**Final Justification:**

The authors answered my questions, and I maintain my high score.

**Quality:**

4

**Strengths And Weaknesses:**

The paper fills a natural gap, by studying a valuable problem (adaptive data analysis) in a relevant setting (growing data). I agree with the authors that growing data is still a sorely understudied setting, even though practical DP applications are actually quite likely to operate under this setting.

The results are significant, since the bounds outperform what could be obtained by composing existing bounds over disjoint static datasets. The proof techniques seem to broadly follow previous work from Jung et al, with some novel time-dependent budget and accuracy variants.

A few things were unclear to me in the evaluation:
- I didn't understand how the round-dependent standard deviation was set in the experiments. Intuitively, I might want to spend more budget on the first queries, so I would expect something like sigma_t proportional to t. Indeed, it should become easier and easier to get good accuracy as data accumulates, at least for statistical queries, where alpha is the absolute counting error normalized by t.
- On a related note, what is the exact time-dependent delta for "Ours-N"? And more generally, what are epsilon and delta in Fig 3, and how can we compare the utility for Ours-N and Ours-U on the same graph when they operate under different privacy guarantees?
- Finally, if I understand correctly, Fig 3 takes a collection of theorems and plots the values under various configurations. But it could be interesting to see an evaluation over a workload of statistical queries over an actual growing dataset. Instead of showing the theoretical number of queries within the error target, could we look at the empirical error distribution for each batch, e.g. with box plots?

Also, regarding the time-dependent guarantees:
- The paper uses a time-dependent delta, but not a time-dependent epsilon. Internally, rho depends on t, but it seems that the zCDP to approximate DP conversion stashes all the time-dependencies into delta. I wonder why and what would be the implications of a time-dependent epsilon.
- The semantics of time-varying delta under fully adaptive composition are left unspecified. The appendix mentions that individual filters could be used, such as the one from Feldman and Zrnic, but does not detail how.
- In the paper, budget only depends on time, but personalized DP supports budget that depends on the data from record themselves. Would individual accounting a la Feldman and Zrnic yield extra savings, e.g. for certain data distributions and queries where most records have low individual sensitivity?

Overall, I found the paper extremely clear and well-written. The results were crisp and motivated. Fig 2 was useful, and the background related work sections were particularly pleasant to follow. As a non-expert, this was a great opportunity to learn about the field -- thank you!

---

> ### Author Rebuttal · Authors · 2025-07-31
>
> We thank you for your thorough review and positive feedback. We're pleased that you recognize the significance of our work in addressing the understudied setting of growing data for adaptive data analysis. We address your questions and comments below. We welcome further feedback, including a potential reconsideration of your score if you feel it's warranted.
>
> > Round-dependent standard deviation in experiments
>
>
>
> We used a constant noise scale $\sigma_t = \sigma$ for all rounds, which results in a constant confidence width $\alpha_t’ = \alpha’$. We chose this approach since:
>
> 1. It simplifies the interpretation of the results: we can report the number of queries that can be answered at a fixed confidence width.
>
> 2. If we considered round-dependent confidence width $\alpha_t’$, we would need to make assumptions about the dependence that an analyst would find tolerable.
>
> > What is the exact time-dependent delta for "Ours-N"? And more generally, what are epsilon and delta in Fig 3, and how can we compare the utility for Ours-N and Ours-U on the same graph when they operate under different privacy guarantees?
>
> The time-dependent $\delta(t)$ for "Ours-N" is defined in Lemma 4.2. The computation involves three steps: (1) the privacy accounting is done using non-uniform zCDP; (2) given $\mathbf{\rho}(t)$ and $\epsilon$, we compute the optimal order of the Renyi divergence $\gamma^\star$; (3) finally, we plug $\mathbf{\rho}(t)$, $\epsilon$ and $\gamma^\star$ into $\psi$ to obtain $\mathbf{\delta}(t)$.
>
> In Figure 3, we follow the approach of Jung et al. (2020) by optimizing over free parameters to obtain tighter bounds. Specifically, we optimize over $\epsilon$, $\sigma$, and $\beta$, with $\delta$ determined by Lemma 4.2. As a result, the privacy guarantee can vary slightly between points, which is acceptable in our context, as our focus is on generalization under adaptation (data reuse) rather than privacy per se. We will clarify this point in Section 4.2 of the paper. It's worth noting that we found the optimal values $\sigma = 0.008$, $\beta = 10^{-5}$, $\epsilon = 0.04$ to be quite stable, varying only beyond the first significant figure.
>
> > Checking what’s plotted in Fig 3, and suggestion to evaluate on an actual growing dataset and analyst.
>
> You're correct that Fig 3 plots theoretical bounds, representing worst-case scenarios for both the data distribution and analyst's strategy. While we agree that case studies with real adaptive analysts could be valuable future work, it’s important to note that:
>
> - Simulation results would be highly dependent on specific choices, potentially limiting their generalizability.
> - Our theoretical analysis is necessary to understand the fundamental limits for any real setting.
>
>  > Why not consider a round-dependent $\epsilon$?
>
> While it's certainly possible to define privacy using an $\epsilon$ that depends on $t$, this approach isn't useful for obtaining generalization bounds, at least with our current proof technique. For example, if we used a time-dependent $\epsilon$, we would encounter issues following equation (15) in our proof of Lemma 3.3, as the sum over $t$:
>
> $$
> \sum_{t = 1}^{n} (e^{\epsilon(t)} - 1) \sum_{\pi \in \Pi^+(\alpha, x)} \mathbb{P}(\Pi = \pi) \frac{1_{t \leq t^\star(\pi)}}{t^\star(\pi)}
> $$
>
> would no longer simplify nicely. This is why we opted for a round-dependent $\delta$ instead.
>
> > Appendix doesn’t detail how to do fully adaptive composition with time-varying delta, e.g. using Feldman and Zrnic (2021).
>
> We did not explain how to use Feldman and Zrnic’s filter, as it uses Renyi DP for accounting, not zCDP, which we used for accounting in Section 4. We would be happy to include it if the reviewer feels this would be helpful. In essence we would integrate Feldman and Zrnic’s Algorithm 3 into our Algorithm 2, with the main change being that $\bar{\rho}$ and $\rho$ become $t$-dependent, and upon termination, the interaction would satisfy $(\alpha, \vec{\rho})$-RDP rather than $\rho$-zCDP.
>
> > Budget only depends on time, but personalized DP supports budget that depends on the data from record themselves. Would individual accounting yield extra savings?
>
> We'd like to clarify that in our paper, the budget actually depends on the _record_ received at round $t$, which is similar to personalized DP. You're correct that individual accounting can yield savings, especially for the “sparse” setting considered by Feldman and Zrnic (2021).

---

> > ### Comment · Reviewer_ow1s · 2025-08-05
> >
> > Thank you for your detailed answer.

---

### Official Review · Reviewer_Cyhf · 2025-07-07

**Clarity:** 4
**Significance:** 4
**Originality:** 4
**Rating:** 5
**Confidence:** 4

**Summary:**

The paper considers application of a differential privacy to address the problem of adaptive data analysis in for a growing dataset.

Specifically, authors consider a sequence of elements $X_1, \ldots X_n$ sampled i.i.d. from a probability distribution $P$ over a universe $\mathcal{X}$. At each timestep $n_0 \leq t \leq n$, the algorithm, having access only to elements $X_1, \ldots X_t$, is supposed to provide an answer to a low-sensitivity query $q$, provided by analyst. (For simplicity, a special case of a low-sensitivity query, is a statistical query, which is specified by a function $q : \mathcal{X} \to [0,1]$. The intended answer to the statistical query is an approximation of $\mathbb{E}_{X \sim \mathcal{P}} q(X)$.) The subsequent queries specified for the mechanism can potentially depend on the answers to the previous queries, and the algorithm aims at answering all those queries approximately correctly, even against an adversarial analyst.

In a static dataset setting, where $X_1, \ldots X_n$ are known to the mechanism as it starts answering the queries, this is extremely well-studied topic. Naively, to achieve simultanous error $\varepsilon$, on all $m$ (adaptively chosen) queries, one would used around $n=O(m/\varepsilon^2)$ samples - $1/\varepsilon^2$ samples for each query. The celebrated line of research stemming from an observation that methods from Differential Privacy could be used to improve the error/sample complexity trade-off: using those methods one can achieve error $\varepsilon$ on $m$ queries using number of samples that grows only as $O(\sqrt{m}/\varepsilon^2)$.

Simultanously, in recent years, a growing body of research addresses providing privacy guarantees using differential privacy under the streaming setting: when the data is not necessairly known to the algorithm at the beginning of the computation, but instead is provided to algorithm one by one.
In this paper authors combine those two prolific lines, providing an answer to a natural, and important question: can differential privacy be used to adaptively answer statistical (or low-sensitivity) queries for a sequence of independent samples from a given distribution, in a streaming scenario.

As in the case of adaptive data analysis in a static data set, they introduce an intermediate notion of "posterior stability", and show that on one hand that any mechanism satisfying differential privacy with particular parameters, also satisfies the posterior stability; then they proceed to showing that the posterior stability in fact implies the accuracy guarantee for an adaptive algorithm operating on a growing data set - i.e. analog of the "transfer theorems" known in the static dataset setting.

Finally, they instantiate this for a specific case of the Gaussian Mechanism, satisfying concrete Differential Privacy guarantees, which by the transfer theorems imply concrete generalization guarantees for the distributional setting. In the appendix they additionally provide a generalization of the framework to "minimization queries", which could be used as a building block of more involved learning algorithms.

**Questions:**

Is there any more interpretable asymptotic guarantee that could be deduced from Theorem 4.4 (or from the framework in general), giving a sense of how the error profile $\alpha_t$ depends on $t$, and the number of queries asked in the batch at time step $t$? Is there a way to interpret formula in the Theorem 4.4?

**Ethical Concerns:**

["NO or VERY MINOR ethics concerns only"]

**Limitations:**

See above.

**Paper Formatting Concerns:**

No concerns.

**Quality:**

4

**Strengths And Weaknesses:**

The paper posits and answers a natural question, combining in a natural way two very prolific lines of research within the differential privacy. It is extremely well written, and provides a rather general theorem translating concrete DP guarantees (that can be achieved thanks to large body of DP primitives, and convenient composability theorems for DP) into statistical generalization guarantees that are desirable as an end goal. The paper is also extremely well written, and provides plots comparing the generalization guarantees of the new mechanism, with attempting to achieve the same by using either trivial methods, or previous DP-based mechanism for static data.

The main weakness of the paper is that even the most concrete instantiation of the framework (Theorem 4.4) still does not provide much intuition about asymptotic behavior of the proposed solution. The formula for error parameter $\alpha'$ surprisingly does not depend on $t$, and is relatively involved, and difficult to interpret. It would be great to see a (potentially worse) asymptotic bound that can be deduced from concrete instantiation of this framework with a specific mechanism, that is more interpretable.

---

> ### Author Rebuttal · Authors · 2025-07-31
>
> Thank you for your thorough review and positive feedback on our paper. We appreciate your recognition of the paper's contribution in combining two lines of research within differential privacy. Below we provide clarification and context for your comments about Theorem 4.4. If you are satisfied, we’d be grateful if you might consider increasing your score.
>
> > Is there any more interpretable asymptotic guarantee that could be deduced from Theorem 4.4 (or from the framework in general), giving a sense of how the error profile $\alpha_t$ depends on $t$, and the number of queries asked in the batch at time step $t$? Is there a way to interpret formula in the Theorem 4.4?
>
> Theorem 4.4 is stated for the Gaussian mechanism with a constant noise scale $\sigma_t = \sigma$. In this case, the confidence width/error profile $\alpha_t’ = \alpha’$ is indeed constant with respect to $t$ (see proof in Appendix C.1). In words, Theorem 4.4 guarantees that the worst-case absolute error across _all_ rounds/queries is bounded by $\alpha’$ with probability $1 - \beta’$.
>
> Our bound can be easily extended to round-dependent $\sigma_t$ (and hence round-dependent $\alpha_t'$) by simply replacing $\sigma$ with $\sigma_t$ in the statement, assuming $\sigma_t \propto \alpha_t$. However, we chose to use a constant $\alpha'$ in our main results for two reasons:
>
> 1. It simplifies the interpretation of the results: we can report the number of queries that can be answered at a fixed confidence width.
>
> 2. If we considered round-dependent confidence width $\alpha_t’$, we would need to make assumptions about the dependence that an analyst would find tolerable.

---

### Note · Authors · 2025-08-15

We sincerely thank all reviewers for their valuable feedback. We believe we've addressed all concerns and comments raised. Key changes we've committed to include:

1. Elaborating on how our analysis differs from Jung et al. (2020)
2. Clarifying our focus on the fully adaptive setting, where the query schedule is not fixed in advance but determined on-the-fly by the analyst

We note that Reviewers Cyhf, ow1s, and KtCZ didn't raise follow-up concerns, suggesting their satisfaction with our responses.

We particularly appreciate the dialogue with Reviewer BVSk. Their key concerns stemmed from an initial misunderstanding about our setting, which we've promised to clarify in revision. Specifically, we'll emphasize that our work provides worst-case bounds for an adaptive query schedule, not a fixed one. This focus explains our highlighted scenario in Fig 3 (substantial initial dataset size) and why some proposed proof enhancements aren't applicable in our context.

Once again, we express our gratitude to all reviewers for their time, thoughtful suggestions, and contributions to improving our work. We look forward to incorporating these insights in our revision.

---

### Decision · Program_Chairs · 2025-09-17

**Decision:**

Accept (poster)

**Comment:**

The reviewers all agreed that this paper was a strong submission. The strengths identified included: it combined two lines of work in the DP literature to fill a natural gap in the literature, it is well-written, it provides a strong set of theoretical results and verified these results empirically. There was a question of novelty with respect to Jung et al., but the authors engagement in the discussion phase helped clarify these, and the authors promised to make this more clear in the final version.

The reviewers were largely positive before the rebuttal phase, and feel that the authors satisfactorily addressed any remaining concerns through their rebuttals, particularly through a lengthy discussion with Reviewer BVSk, which clarified details of the problem setting.